# Systems immunology integrates the complex endotypes of recessive dystrophic epidermolysis bullosa

Nell Hirt [1,11], Enzo Manchon [1,11], Qian Chen[2], Clara Delaroque[3], Aurelien Corneau[4], Patrice Hemon [5], Safaa Saker-Delye[6], Pauline Bataille[7], Jean-David Bouaziz[1,7], Emmanuelle Bourrat [7], Alain Hovnanian [8], Helene Le Buanec[1], Fawzi Aoudjit[9], Hicham El Costa [10], Nabila Jabrane-Ferrat [10,12] & Reem Al-Daccak [1,12] ✉

Endotypes are characterized by the immunological, inflammatory, metabolic, and remodelling pathways that explain the mechanisms underlying the clinical presentation (phenotype) of a disease. Recessive dystrophic epidermolysis bullosa (RDEB) is a severe blistering disease caused by *COL7A1* pathogenic variants. Although underscored by animal studies, the endotypes of human RDEB are poorly understood. To fill this gap, we apply systems immunology approaches using single-cell high-dimensional techniques to capture the signature of peripheral immune cells and the diversity of metabolic profiles in RDEB adults, sampled outside of any opportunistic infection and active cancer. Our study, demonstrates the particular inflammation and immunity characteristics of RDEB adults, with activated / effector T and dysfunctional natural killer cell signatures, concomitant with an overall pro-inflammatory lipid signature. Artificial intelligence prediction models and principal component analysis stress that RDEB is not solely confined to cutaneous issues but has complex systemic endotypes marked by immune dysregulation and hyperinflammation. By characterising the phenotype-endotype association in RDEB adults, our study lays the groundwork for translational interventions that could by lessening inflammation, alleviate the everlasting suffering of RDEB patients, while awaiting curative genetic therapies.

Recessive Dystrophic Epidermolysis Bullosa (RDEB), an inherited rare skin disorder with no current cure, is caused by pathogenic variants in the human *COL7A1* gene encoding type VII collagen (Coll-VII)[1]. Coll-VII forms anchoring fibrils, which are crucial for dermal-epidermal integrity. The loss of function, reduction or absence of anchoring fibrils lead to chronic skin and mucous membrane injury with progressive multiorgan effects. Dermal fibrosis and aggressive squamous cell carcinoma (SCC) are recurrent invalidating complications of RDEB adults. Besides causative genetic mutations, similar to other monogenetic diseases, both local and systemic manifestations of RDEB may involve other cellular and molecular mechanisms[2].

The aforementioned notion has been advanced through various animal models recapitulating, more or less, the clinical, immunohistochemical, and ultrastructural characteristics of human RDEB[3–6]. The *COL7A1*-hypomorphic RDEB mouse model revealed that the absence of the innate immune activator, cochlin, in lymphoid conduits is associated with innate immune cell dysfunction and increased bacterial colonization[7]. Analyses of skin wounds, dressings, and sera from RDEB patients have also revealed altered abundance and activity of immune cells and immune mediators[6,8–14]. Noteworthy, side-by-side CRISPR/Cas9-mediated knockout of *Col7a1* in NSG mice, lacking T, B, and NK cells, led to less severe RDEB and prolonged survival compared to

immunocompetent C57Bl/6 mice[5]. Collectively, these observations bring into question the landscape of RDEB beyond skin and mucosa problems.

Endotypes are characterized by the immunological, inflammatory, metabolic, and remodelling pathways that explain the mechanisms underlying a clinical presentation (phenotype) of a disease. Knowledge regarding RDEB endotypes is mainly stipulated within mice models, and from scattered studies in human RDEB injured skin. The beneficial effect of losartan treatment of RDEB mice was attributed to reduced levels of circulating inflammatory cytokines resulting in lower tissue inflammation and milder disease[8]. Similarly, the main benefit of stem cell-based therapies is lessening tissue inflammation and improving the healing of RDEB wounds[15–17]. In this work, we apply systems immunology approaches using in-depth immune phenotyping by mass cytometry and large-scale profiling of the energy and lipid metabolism in peripheral blood clinical samples, and subject our findings to machine learning algorithms and artificial intelligence. Here, we demonstrate the particular inflammation and immunity characteristics of RDEB adults, with activated / effector T and dysfunctional natural killer cell signatures, concomitant with an overall pro-inflammatory lipid signature. Collectively, our data substantiate and stress that RDEB is not solely confined to cutaneous issues but has complex systemic endotypes marked by immune dysregulation and hyperinflammation.

## Results

### RDEB patient's clinical characteristics

Our cohort included twelve non-end-stage RDEB adults (8 females, 4 males) diagnosed at birth with different *COL7A1* pathologic variants and levels of disease severity, which inevitably worsened with age to a degree highly related to their initial one (Supplementary Table 1 and 2). All patients are classified according to their IscorEB (Instrument for Scoring Clinical Outcome of Research for Epidermolysis Bullosa), the scoring system commonly used in international clinical trials[17] and by the "French National Reference Centre for Rare Diseases of the skin and mucous membranes of genetic origin (MAGEC)" which manages the RDEB cohorts. IscorEB combines clinical manifestations grouped in five domains (skin, mucosa, organ involvement, laboratory abnormalities, and complications and procedures; maximum score 138) and patient-derived items (pain, itch, functional limitations, sleep, mood, and effect on daily and leisurely activities; maximum score 120), resulting in a total maximum score 258[18]. At enrolment, the RDEB adults presented a body mass index <23 indicating common malnutrition status, a percentage of wound area ranging between 20–80%, and a mean IscorEB of 78±25. All patients were sampled outside of any opportunistic infection and active cancer. Ten out of the 12 patients presented relatively homogenous IscorEB. The other two are a 52 year-old patient suffering from localized RDEB with 20% wound area and the lowest IscorEB at 33, and a 25 year-old patient with severe generalized RDEB, 55% skin wounding, and the most severe clinical manifestations reflected in the highest IscorEB at 133. Since the two extremes did not affect the global IscorEB, we have decided to include all 12 patients (mean age = 34.6 ± 8.8) in this study. Nine age-matched blood donors (5 females, 4 males; mean age = 30 ± 5) were used as healthy controls (HC).

### RDEB patient's immune landscape

Knowledge concerning RDEB patients' peripheral immune cell populations is limited to a few reports[19–22]. To advance the understanding on the immunology of RDEB, we conducted a comprehensive, systematic interrogation of peripheral blood leucocytes in RDEB adults collected outside of any clinical superinfection or active cancer. We applied the single-cell mass cytometry (CyTOF) method to monitor CD45+-circulating immune cells in whole blood samples from patients and healthy controls (Supplementary Fig. 1a). A large panel of validated lineage-

specific metal-tagged antibodies was applied (Supplementary Table 3), to allow a thorough investigation of the diversity of immune cell populations in HC and RDEB groups.

Here, we used uniform manifold approximation and projection (UMAP) as it provides a faithful representation of the relative distances of cell clusters in high-dimensional space. Dimensional reduction using UMAP coupled to an unbiased clustering algorithm, PhenoGraph, reflected dynamic changes in the proportions of various immune cell populations (Fig. 1a). Eleven distinctive clusters were identified and manually annotated according to cell surface expression patterns of lineage-specific markers CD3, CD4, CD8, CD56, CD161 and TCRγδ for T cells, CD20 and CD19 for B cells, CD56 and CD16 for natural killer (NK) cells, CD14, HLA-DR, CD66b, CD11b, CD11c, CD294, CD33, CD38 and CD123 for myeloid cells (Supplementary Fig. 1b). Illustration of UMAP density plots revealed substantial differences in major innate and adaptive immune cell populations of RDEB adults compared to HC, particularly in neutrophils, monocytes, dendritic cells, and lymphocytes (T, B, and NK cells) (Fig. 1a right two panels). When evaluating relative frequencies of the broadly defined cell populations, the most notable changes in RDEB adults were elevated frequency of neutrophils, with a reduction in basophils/eosinophils and monocytes, and a less pronounced decrease in dendritic cells and lymphocytes (Fig. 1b). Given that the high abundance of neutrophils could be a major driver of the shifts in the relative frequency of other immune cell populations, for a more realistic view we determined the absolute cell counts for each population. The absolute counts confirmed the abundance of neutrophils in RDEB patients and the decreased relative frequency of dendritic cells (Fig. 1b). Yet, monocytes were present at significantly higher absolute numbers in RDEB patients whereas lymphocytes and basophils/eosinophils had similar counts as HC (Fig. 1b).

Since skin wounding is a main actor in RDEB pathogenic chain, to further comfort these peripheral observations, we applied imaging mass cytometry (IMC) and an optimized protocol[23] using a panel of 30 metal-conjugated antibodies (Supplementary Table 4) to identify the immune cell infiltrates in formalin-fixed paraffin-embedded (FFPE) tissue sections from 2 RDEB adults skin biopsies. Compared to tissue sections from HC, skin lesions from both RDEB patients showed an overall drastically distressed structure and considerably elevated infiltrates of various CD45+ immune cell populations (Supplementary Fig. 2a and Fig. 1c). Using the opt-SNE dimensionality reduction tool coupled to PhenoGraph-based meta clustering, we visualized dynamic changes in the proportions of various immune cell populations reflecting those observed with peripheral immune cells from RDEB adults. Overall, we identified 10 immune cell populations using marker expression patterns (Supplementary Fig. 2b), and found that RDEB skin lesions, in contrast to healthy tissue section, were marked by the presence of neutrophils, B cells, and plasma cells, as well as elevated numbers of activated CD4+ and CD8+ T, regulatory T cells, monocytes/DC, and macrophages (Fig.1d). Taken together, these results show that RDEB adults display different proportions of various immune cell populations, both in the periphery and the skin.

### PBMC profiling of RDEB patients

To provide further insight into the major adaptive and innate immune cell populations of RDEB adults, we performed an in-depth analysis of their peripheral blood mononuclear cells (PBMC). Dimensionality reduction via UMAP coupled to PhenoGraph-based meta clustering (Fig. 2a and Supplementary Table 3) reflected several changes in the frequency of major populations of PBMC in patients compared to HC (Fig. 2b, Supplementary Fig. 3a and Supplementary Table 5). Volcano plot (Supplementary Fig. 3b) revealed increased percentages of effector and central memory (CM) CD4+ (4% vs 0.1% and 19.3% vs 13%, respectively) and CD8+ (5% vs 1.6% and 8.0% vs 5.2%, respectively) T cell subsets, as well as of CD14+CD16+ intermediate monocytes in RDEB

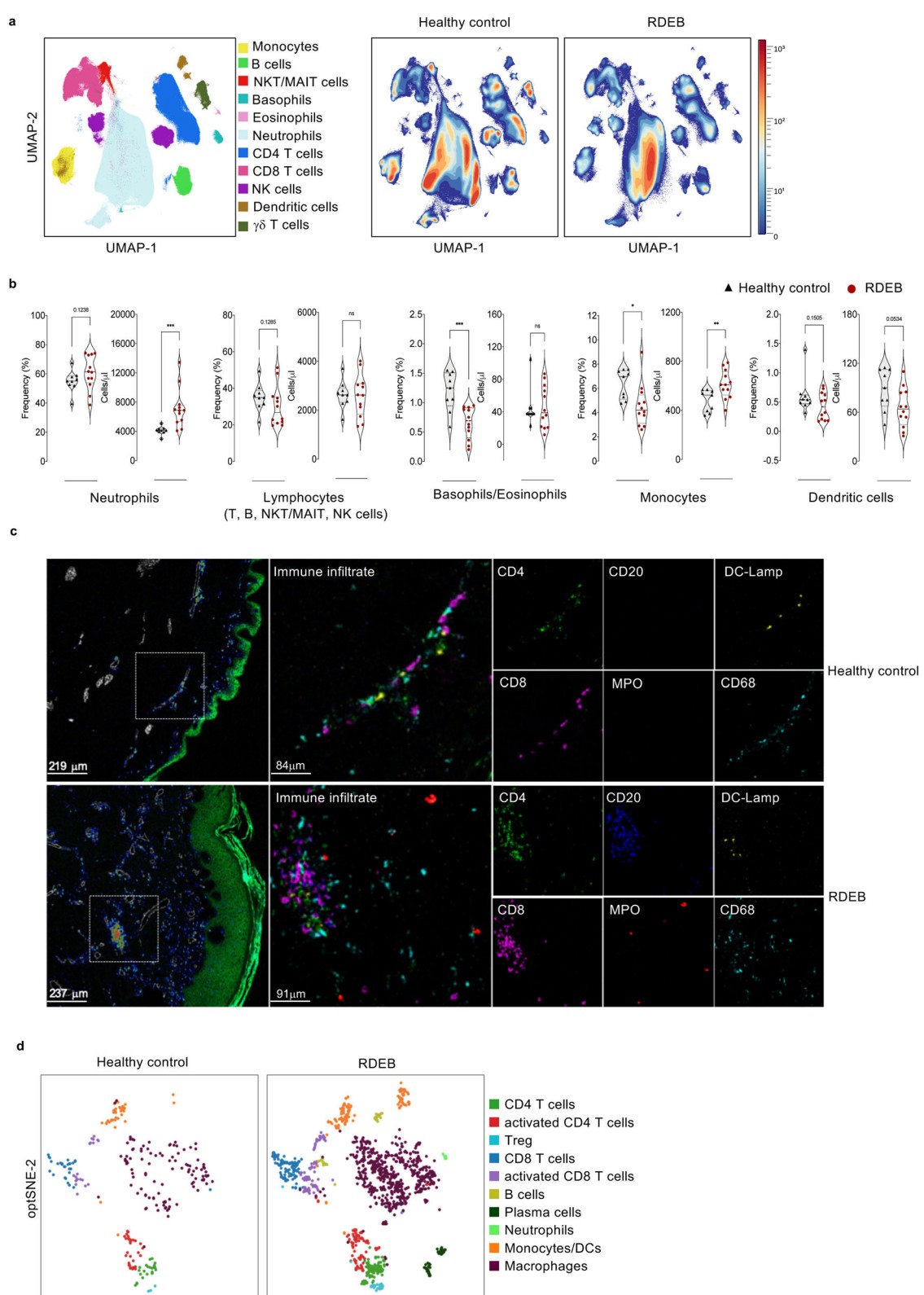

adults compared to HC (Fig. 2b). In contrast, γδ T cells, both CD56[bright] and CD56[dim] NK cell subsets, and CD14[+]CD16[-] classical monocytes were decreased in patients (2.6% vs 6.5%, 0.4% vs 1.1%, 4.3% vs 8.4%, 9.0% vs 14.6%) (Fig. 2b). Almost a mirror image was obtained for absolute cell counts of T and non-T cell subsets indicating concordant changes (Fig. 2c). While classical monocytes were less abundant in patients, they were present at similar absolute counts as HC. We did not observe any evident change in the percentages or absolute cell counts of either

naïve CD4[+], effector memory (EM) CD4[+] and CD8[+] T cell subsets, natural killer T (NKT)/mucosal-associated invariant T cells (MAIT) and NK CD56[dim]CD57[+] cells, B cells, and CD14[-]CD16[+] non-classical monocytes (Fig. 2c). Conclusively, the principal component analysis (PCA) based on the distribution of immune cell populations highlighted discriminative results between patients and healthy individuals, further emphasizing the particular distribution of these populations in RDEB adults (Fig. 2d).

**Fig. 1 | Immune landscape of RDEB adults. a** UMAP visualizing PhenoGraph-obtained clusters of major immune cell populations from a total of 21 whole blood samples ($105.10^3$ cells/sample). UMAP comparing the distribution of major immune cell populations in 9 healthy controls and 12 RDEB adults. Colours vary according to cell abundance density. **b** Violin plots comparing the frequency (%) and the absolute counts (cells/μl) of indicated immune cell populations of 9 healthy controls (black triangles) and 12 RDEB adults (red circles) with median values presented by solid black line. Source data are provided as a Source Data file. Statistical analyses were performed with two-sided unpaired $t$-test. Asterisks represent the significant differences (*$p < 0.05$, **$p < 0.01$, ***$p < 0.001$). **c** Variability in single-cell distributions across Healthy and RDEB adult Formalin-fixed paraffin-embedded skin slices.

Representative (three independent experiments) images of antibody staining and corresponding single-cell segmented images across histological subgroups. Displayed labelling patterns for each reconstructed image. Merge: panKeratin(green)/SMA (grey)/immune cell subsets (mixed colours). Enlarged images depicting the immune cell subsets present within the dotted square. Immune cell subsets depicted include CD4 T cells: CD4 (green), CD8 T cells: CD8 (magenta), Monocytes/Macrophages: CD68 (cyan), B cells: CD20 (bleu), Dendritic cells: DC-Lamp (yellow), neutrophils: MPO (red). **d** Opt-SNE visualizing PhenoGraph-obtained clusters of CD45$^+$ immune cells from healthy (HC, $n = 2$) and RDEB adult ($n = 2$) skin biopsies. Each dot represents a single cell.

## RDEB CD4$^+$ and CD8$^+$ T cells homoeostasis

Given the critical role of CD4$^+$ and CD8$^+$ T cells in regulating health and disease and the observed high levels of activated T cells in RDEB tissue sections, even subtle changes in the peripheral ratios of their respective subsets might affect the fate of the overall immune response. To obtain an unbiased picture of how all markers are behaving and circumvent the risk of missing critical information, we applied Flow Self-Organizing Maps, FlowSOM, to obtain an unsupervised T cell clustering. For controls and patients, data from CD4$^+$ and CD8$^+$ T cells were exported and respectively, concatenated in a unique matrix then subjected to dimensionality reduction algorithm opt-SNE and FlowSOM-based meta-clustering (Supplementary Table 3). Based on the expression of surface markers, we identified 12 clusters for CD4$^+$ and CD8$^+$ T cells (Supplementary Figs. 4a and 5a, respectively) and visualized various cell subsets (Supplementary Table 5) with their respective percentages (Fig. 3a, b).

Volcano plot (Supplementary Fig. 4b) showed that the percentage (Fig. 3a) of effector CD4$^+$ cells, expressing or not the terminal differentiation/senescence marker CD57, and CM Th1 CD4$^+$ subsets were significantly increased in RDEB patients while naïve CD4$^+$ T and Treg cells were significantly decreased. The absolute cell numbers (Fig. 3c) were in concordance with the percentages in most subsets suggesting that RDEB adults have an overabundance/overproduction of effector CD4$^+$ T cells at the expense of naïve and regulatory T cells. While the percentage of non-classical Th1, the CM Th1 CCR6$^+$CD161$^+$ subset often related to pathologic inflammation, was slightly higher in RDEB patients compared to HC (9.2% vs 8.4%, respectively), the absolute counts demonstrated that their numbers were significantly increased in patients (Fig. 3c).

RDEB patients also showed an overabundance of effector CD8$^+$ T and MAIT cell subsets, expressing or not CD57, as evidenced by increases in both their percentage and absolute numbers (Fig. 3b, d and Supplementary Fig. 5c). In contrast, RDEB adults showed decreased percentage, but not absolute counts, of CD57$^+$ EM CD8$^+$ T cells and NKT/MAIT cells, implicated in immune response against infectious agents (Fig. 3b and Supplementary Fig. 5c). The CD57$^-$ EM CD8$^+$ T cells were present at higher percentages and numbers in RDEB patients compared to HC (Fig. 3b, d). The absolute cell numbers mirrored the percentages of the other cell subsets suggesting that RDEB adults have also an overabundance/overproduction of effector CD8$^+$ T cells compared to naïve cells.

We then delineated the proliferative capacity of CD4$^+$ and CD8$^+$ T cell compartments in RDEB patients in the absence or presence of stimulation. Four-days culturing did not trigger any change in the proliferation or replication index of either CD4$^+$ and CD8$^+$ T cells (Supplementary Fig. 6). Nonetheless, RDEB CD4$^+$ (Fig. 3e) and CD8$^+$ (Fig. 3f) cells were able to proliferate in response to 4-days of stimulation with ImmunoCult CD3/CD28/CD2 antibody complexes, although with slightly lower capacity than HC T cells (CD4$^+$ replication index 11.7 vs 16.8, proliferation index 3.1 vs 3.6, respectively; CD8$^+$ replication index 13.1 vs 18.0, proliferation index 3.3 vs 3.7, respectively).

Overall, these results show that RDEB nurtures irregular ratio of naïve to effector T cells in both CD4$^+$ or CD8$^+$ compartments likely biased towards an effector profile.

## Immune signature and activity of RDEB NK cells

Given their important role in immune responses to viral infections and tumours, we submitted RDEB or HC NK cells to dimensionality reduction algorithm, opt-SNE then to unsupervised FlowSOM-based meta-clustering. Based on the expression of 23 surface markers (Supplementary Table 3), we identified 4 clusters of NK cells (Supplementary Fig. 7a and Supplementary Table 5) along with their respective percentages (Fig. 4a and Supplementary Fig. 7b). While the percentage of NK CD56$^{dim}$CD57$^+$, probably presenting memory NK cells, were increased in RDEB patients compared to HC, both CD56$^{bright}$ and CD56$^{dim}$CD57$^-$ NK cells were decreased. Although the absolute number of CD56$^{bright}$ and CD56$^{dim}$CD57$^-$ NK cells concords with their respective abundance, NK CD56$^{dim}$CD57$^+$ cells were present at similar numbers in RDEB patients and HC (Fig. 4b). These results demonstrate that the disturbed proportion of immune cell subsets in RDEB patients is not limited to T cells, but is also reflected in the distribution of NK cell subsets.

Since phenotypic changes in the NK cell compartment might frame their activity, we assessed the functional status, cytotoxicity and cytokine production, of RDEB NK cells. To assess cytotoxicity, IL-15-primed PBMC were cultured with K562 NK-sensitive target cells, and the expression of CD107a degranulation marker by NK cells and the specific lysis of target cells were determined. Compared to HC, RDEB NK cells showed decreased degranulation capacity evidenced by a significantly lower expression of CD107a, associated with a drastic decrease in their cytotoxic function (Fig. 4c). RDEB NK cells also produced less IFNγ and TNF cytokines upon stimulation with PMA/Ionomycin as evidenced by the percentage of positive cells and the related geometric means (Fig. 4d).

To provide a meaningful perception to these findings, we analysed the expression of critical NK cells activating and inhibitory receptors, and the expression of exhaustion- and terminal differentiation/senescence-related markers[24-26]. RDEB NK cells displayed significantly reduced percentage and expression of the activating NKG2D receptor along with significantly higher percentage of the inhibitory KIR2D receptor (Fig. 4e and Supplementary Fig. 7c). NK cells from RDEB patients also showed higher percentage of PD-1- and LAG-3-positive cells, higher expression of senescence-related inhibitory NKG2A receptor, and tended to express higher levels of the activating NKG2C receptor often associated with chronic viral infection (Fig. 4e and Supplementary Fig. 7c). Taken together, these observations demonstrate that RDEB patients likely nurture defective NK cells, marked by a particular phenotype and diminished global activity.

## Single-cell metabolic profiling of RDEB T and NK cells

The differentiation and effector functions of T and NK cells are intimately linked to their metabolism, which is often shaped by the surrounding microenvironment[27-29]. To provide further insights into the

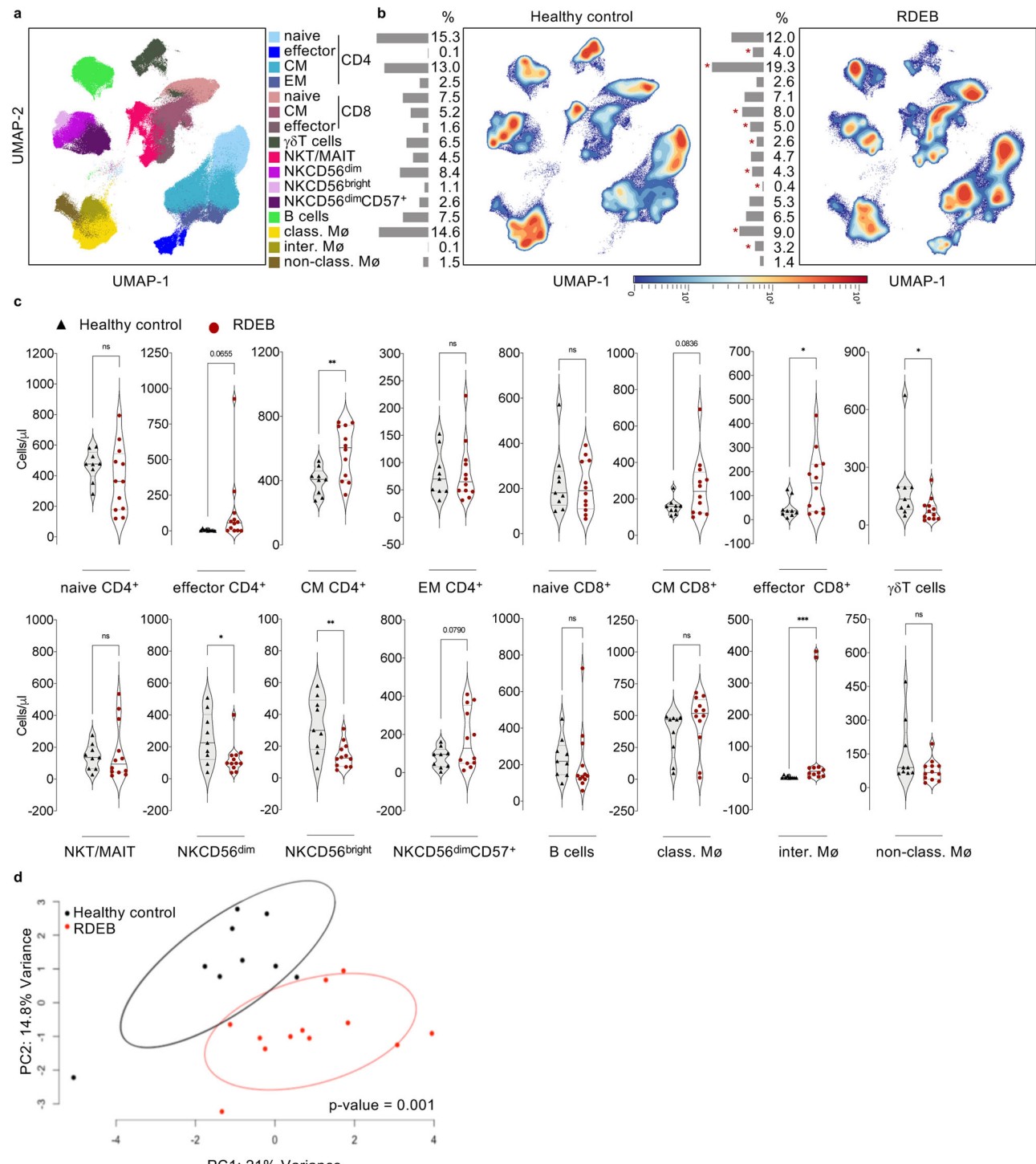

**Fig. 2 | Unsupervised PBMC profiling of RDEB adults. a** UMAP visualizing PhenoGraph-obtained clusters of major innate and adaptive immune cell populations from a total of 21 PBMC samples (25.10³ cells/sample). **b** UMAP comparing the distribution of adaptive immune cell populations in 9 healthy controls and 12 RDEB adults. Colours vary according to cell abundance density. The frequency (%) of each cell population is presented as bar graphs on the left side of the corresponding UMAP. **c** Violin plots comparing the absolute counts (cells/μl) of indicated immune cell populations in 9 healthy controls (black triangles) and 12 RDEB adults (red circles) with median values presented by solid black line. Source data are provided as a Source Data file. Statistical analyses were performed with two-sided unpaired *t*-test. Asterisks represent the significant differences (*$p < 0.05$, **$p < 0.01$, ***$p < 0.001$). **d** PCA representing mass cytometry data from 9 healthy controls (black circles) compared to that from 12 RDEB adults (red circles), PC1: 14.8%, PC2: 21.0%. Clustering significance was determined using Permutational multivariate analysis of variance (PERMANOVA).

observed RDEB T and NK signature, we looked into the metabolic contexture of these cells in a group of patients ($n = 6$) presenting the diverse clinical picture of RDEB adults using SCENITH coupled with multiparametric flow cytometry. This method relies on the analysis of protein translation by puromycin incorporation as surrogate of metabolic activity and the use of specific inhibitors allowing the assessment of glucose dependence, mitochondrial dependence (mitochondrial oxidative phosphorylation), glycolytic capacity, and

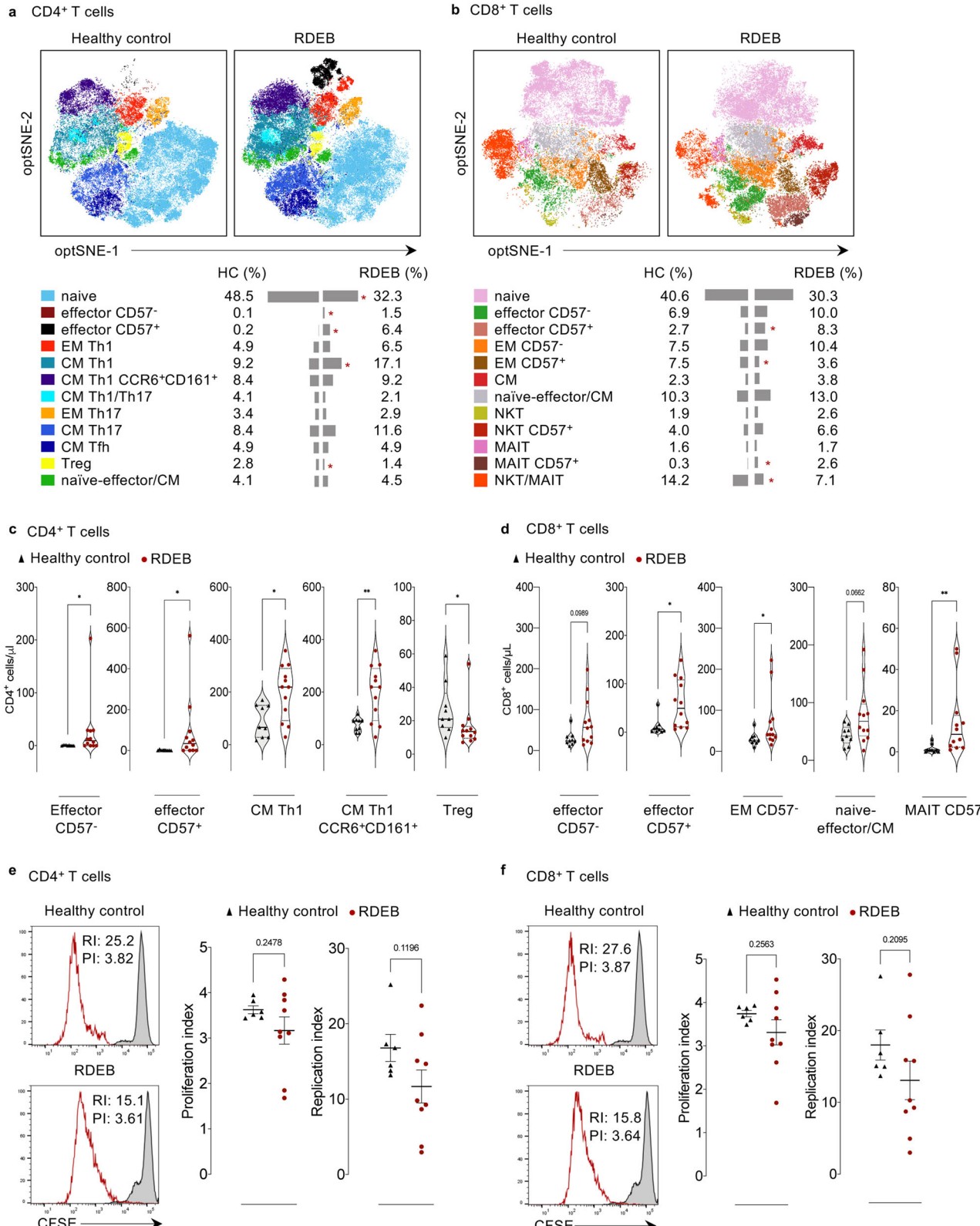

**Fig. 3 | RDEB T cells nurse distinct signature and activity. a, b** opt-SNE from healthy controls ($n = 9$) and RDEB adults ($n = 12$) clustered and coloured by main CD4⁺ and CD8⁺ subsets with their respective abundances (%) by PhenoGraph clustering. Clustering performed with $8.10^3$ CD4⁺ cells/sample and $4.10^3$ CD8⁺ cells/sample. **c, d** Violin plots comparing the absolute counts (cells/μl) of 9 healthy controls (black triangles) and 12 RDEB adults (red circles) CD4⁺ and CD8⁺ T cell subsets with median values presented by solid black line. Source data are provided as a Source Data file. **e, f** CD2/CD3/CD28-PBMC-induced proliferation gated on

CD4⁺ and CD8⁺ cells. Representative histograms of proliferation measured as loss (red line) of initial CFSE labelling (black line) are presented. Replication and Proliferation index (RI, PI respectively) are presented as scatter plots. Data are presented as mean values ± SEM from 6 healthy controls and 9 RDEB adults. Healthy controls (black triangles) and RDEB adults (red circles). Statistical analysis is performed with two-sided unpaired *t*-test. Asterisks represent the significant differences between healthy controls and RDEB adults (*$p < 0.05$, **$p < 0.01$). Source data are provided as a Source Data file.

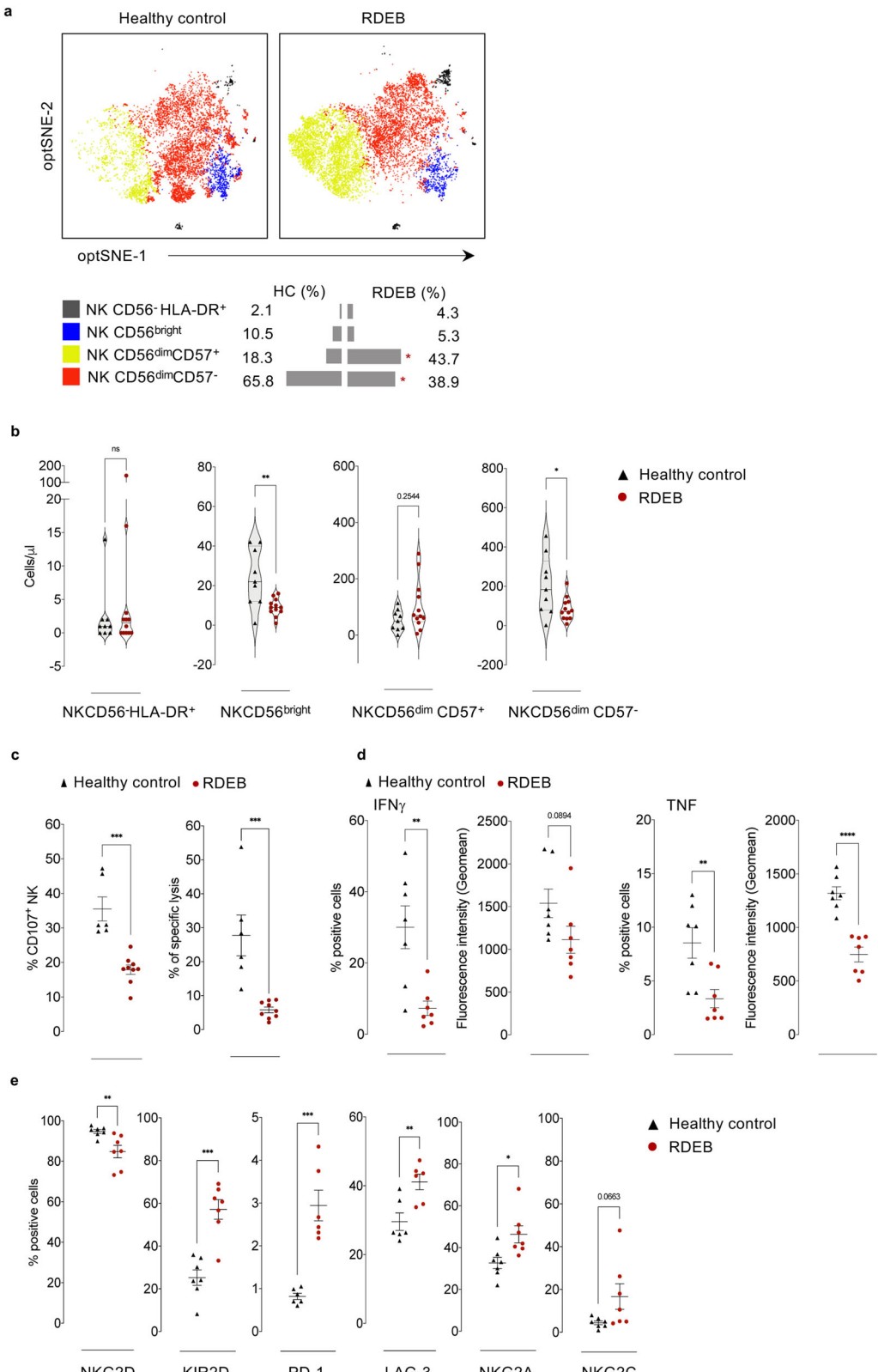

fatty acid and amino acid oxidation (FAAO) (Fig. 5a). Manual gating on CD3, CD4, CD8, CD56 expressing cells (Supplementary Fig. 8) combined with SCENITH allowed us to identify and analyse the cellular metabolism of T and NK cells.

CD4$^+$ T cells and to a lesser extent CD8$^+$ T cells from RDEB patients showed higher translational level than those from HC (Fig. 5b, c), which indicates high metabolic activity supporting a potential activated/

inflammatory pathologic state. RDEB T cells showed a higher degree of glycolytic and FAAO capacity indicating that these cells increase their compensatory capacity and exploit alternative pathways under challenging pathologic environment to meet their energy demands. The high compensatory capacity of T cells was associated with a decrease in glucose and mitochondrial dependence (Fig. 5b, c). Similar to T cells, the global metabolic profiling of RDEB NK cells was different from that

**Fig. 4 | NK cells signature and activity in RDEB adults. a** opt-SNE from healthy controls ($n = 9$) and RDEB adults ($n = 12$) clustered and coloured by NK cell subsets with their respective abundance (%) determined by FlowSOM-based clustering. **b** Violin plots comparing the absolute counts (cells/μl) of NK cell subsets. **c** Degranulation of NK cells assessed by the expression of CD107a marker (left panel) and specific lysis (%) of K562 cells (right panel). Results are presented in scattered plots as mean values ± SEM from 6 healthy controls and 9 RDEB adults. **d** PMA/Ionomycin-PBMC-induced IFNγ and TNF production by NK cells presented as % positive cells with their respective geometric mean fluorescence. Results are presented in scattered plots as mean values ± SEM from 7 healthy controls and 7

RDEB adults. **e** The expression of activating and inhibitory NK cells receptors, and the exhaustion- and senescence-related markers. Results are presented in scattered plots as mean values ± SEM from 7 healthy controls and 7 RDEB adults (NKG2D, KIR2D, NKG2C, NKG2A), mean values ± SEM from 6 healthy controls and 6 RDEB adults (PD-L1 and LAG-3). Healthy controls (black triangles) and RDEB adults (red circles). Statistical analyses were performed with two-sided unpaired $t$-test. Asterisks represent the significant differences between healthy controls and RDEB adults (*$p < 0.05$, **$p < 0.01$, ***$p < 0.001$, ****$p < 0.0001$). Source data are provided as a Source Data file.

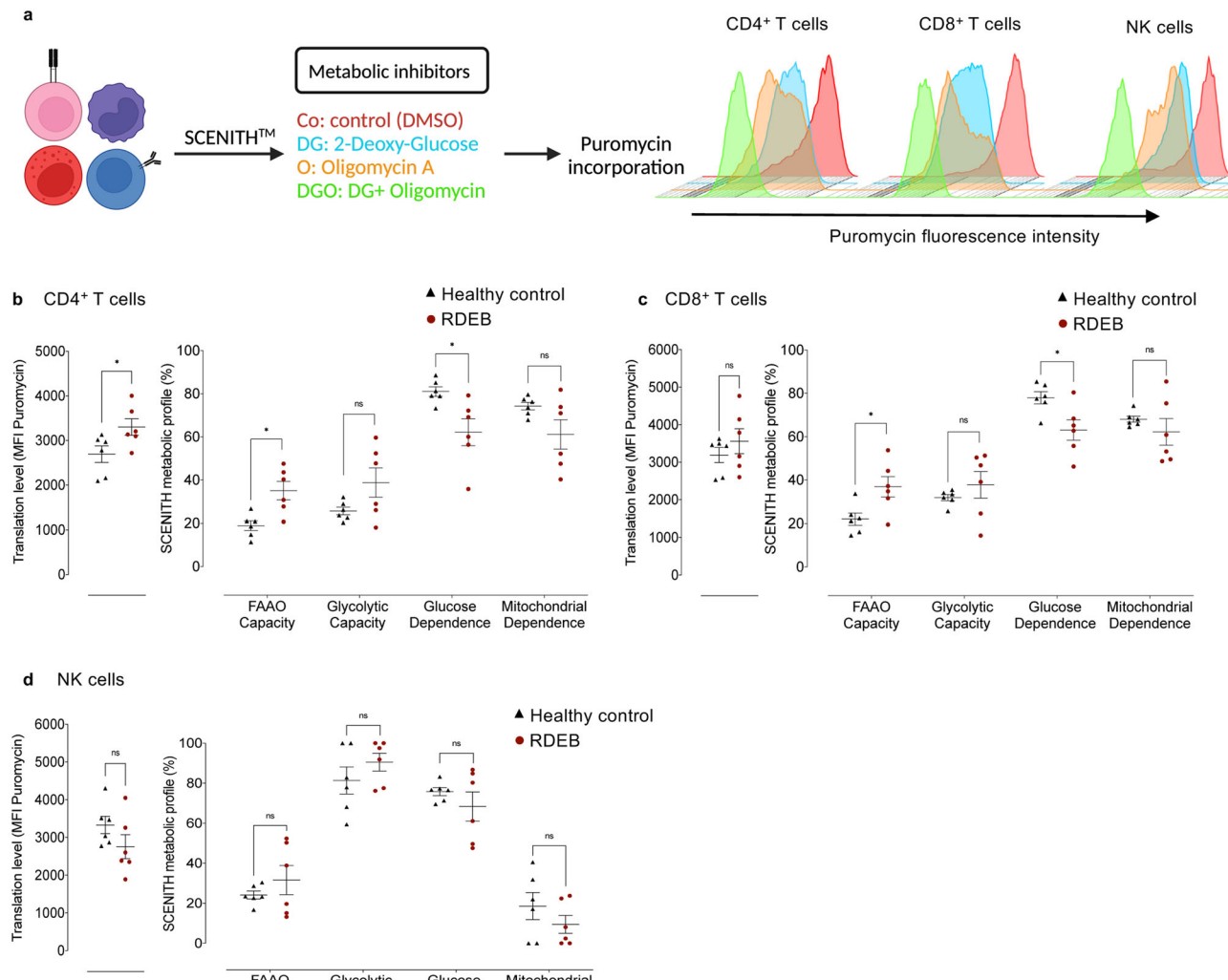

**Fig. 5 | Single-cell metabolic profiling of RDEB T and NK cells. a** Schematic presentation of single-cell energetic metabolism by profiling translation inhibition (SCENITH). Visualization of protein synthesis (PS) after puromycin incorporation by flow cytometry. Histograms display the level of PS in control (Co) and in the presence of specific metabolic inhibitors (DG, O or DGO). **b**–**d** Single-cell metabolic profile presented as scatter plots displaying the translation level, Fatty acid Amino acid oxidation (FAAO), glycolytic capacity, glucose dependence and mitochondrial

dependence in CD4⁺, CD8⁺ cells and NK cells. Healthy controls ($n = 6$, black triangles) and RDEB adults ($n = 6$, red circles). All results are presented in scattered plots as mean values from 6 independent donors ± SEM. Statistical analyses were performed with two-sided unpaired $t$-test. Asterisks represent significant differences between healthy controls and RDEB adults (*$p < 0.05$, **$p < 0.01$). Source data are provided as a Source Data file.

of HC. The glycolytic and FAAO capacity of NK cells in RDEB patients were comparable to HC. Although not statistically significant, RDEB NK cells showed a tendency to reduce their glucose and mitochondrial dependence as well as translation levels (Fig. 5d). Compared to the currently recognized NK cell metabolism[30], such metabolic alterations concord with the overall signature of NK cells from RDEB patients. Altogether, our findings provide a possible basis for the RDEB active/

effector T and dysfunctional NK cell signatures, emphasizing the pathologic immune ground of the disease.

## RDEB patients' lipid metabolism
Lipids, in particular bioactive mediators, act as wildfire-starters of inflammatory processes and have been linked to several chronic diseases, including cancer and autoimmune disorders[31]. The observed

changes in patients' energetic metabolism, particularly the increase in FAAO, prompted us to evaluate whether these immune cells rely on alternative metabolic substrates i.e. lipids. Within these notions, we profiled two main classes of lipids, neutral lipids (NL) and bioactive lipid mediators namely eicosanoids that play important roles in inflammation and immune modulation.

Serum samples from RDEB patients and HC were subjected to large-scale quantitative shotgun lipidomic coupled to mass spectrometry (LC-MS/MS) analyses of NL and LC-MS/MS oxylipins including eicosanoids. Compared to HC, RDEB patient samples revealed a significant decrease in the total amounts of NLs (Supplementary Fig. 9a). However, only minor differences were observed in the relative distribution of the three major subgroups of NLs (cholesterol, cholesterol ester (CE) and triacylglycerols (TG)) (Supplementary Fig. 9b).

We next quantified thirty-five individual eicosanoids and categorized them based on their precursor molecules. Eicosanoids are derived from polyunsaturated fatty acids (PUFAs) (Fig. 6a). Prostaglandins (PGs), Thromboxanes (TXs) and Leukotrienes (LTs) are well-known mediators of inflammation while Lipoxins and Epoxyeicosatrienoic acids (EETs) both acts as pro-resolving. Twelve of the 35 eicosanoids were significantly altered in RDEB patients compared to HC (Fig. 6b). Arachidonic Acid (AA) metabolites TXB2, PGF2a, PGE2, LTB4, 5-HETE, 12-HETE, and 15-HETE were increased in patients whereas PGD2 and LXA4 were decreased. Linoleic acid (LA) metabolites, 9-HODE and 13-HODE, were decreased in patients, while the longer docosahexaenoic acid (DHA) metabolite 14-HDoHE was increased (Fig. 6c–e). Although patients showed a range of metabolic heterogeneity the overall, lipid signature is marked by high levels of pro-inflammatory metabolites concomitant with a considerable decrease of anti-inflammatory metabolites. These data further highlight the inflammatory immune profile of RDEB adults.

### Lipid and immunometabolism interconnectivity in RDEB

To further map the lipid rewiring in RDEB patients, we performed computational Ingenuity Pathway Analysis (IPA). IPA revealed an important upregulation of (1) various signalling mediators, including eicosanoids, TREM1, and IL17; (2) several signalling pathways involved in the regulation of inflammation, immune response, and tumour progression including, oxytocin, ID1, activin inhibition, MSP-RON, IL33, and Th17-activation signalling pathways; and (3) MIF-mediated glucocorticoid and innate immunity regulation (Fig. 7a). Pathway profiling was concordant with the predicted upstream regulators including zymosan and AA as well as various pro-inflammatory cytokines such as TNF, IL1, and MIF, all having eicosanoid as target molecules (Supplementary Table 6). Within the observed RDEB adults' immune-metabolic signature, we evaluated common pro-inflammatory cytokines (IFNγ, IL17a, IL6, TNF), and the canonical anti-inflammatory IL10 cytokines in sera from RDEB patients. In accordance with previous reports[6,8,9,32,33], the mean values ± SEM of all evaluated cytokines were significantly increased in patients compared to controls; IFNγ (1184 ± 371 vs 157 ± 50 pg/ml), IL17a (54 ± 15 vs 16 ± 3 pg/ml), IL6 (121 ± 45 vs 12 ± 2 pg/ml), TNF (650 ± 167 vs 219 ± 47 pg/ml), IL10 (557 ± 161 vs 93 ± 22 pg/ml) (Fig. 7b). In addition to the recognized pro-inflammatory cytokines (IFNγ, IL17a, IL6, TNF), the concomitant increase in the canonical anti-inflammatory IL10 can also be a feature of hyperinflammation[34]. This cytokine signature is in line with the observed immunometabolic signature of RDEB adults and corroborates the current notion of systemic inflammation in RDEB[35,36]. The IPA interrogation strengthens the link of RDEB adult lipid signature not only with metabolism and immune inflammation, but also with dermatological and infectious injuries, and cancer, which are integrated aspects of RDEB (Fig. 7c).

### Deep learning prediction models for RDEB

In light of the aforementioned findings that underscore the wide spectrum of immunometabolism in RDEB adults, we assessed and visualized a predictive signature using various parameters, including cytokine levels, lipid profiles, and absolute counts of circulating immune cells, collectively referred to as the Inflammation Immunity Score (IIS). In addition to IIS, we included the IscorEB, % of wound area, and incidence of SCC. Correlation matrix comforted the IPA associations between certain immune cell subsets, lipid mediators, and cytokines (Supplementary Fig. 10). The IIS dataset did not enable successive prediction of the IscorEB, % of wound area, or incidence of SCC as evidenced by an area under the curve (AUC) close to 0.17, 0.44 and 0.45 respectively, indicating an equal number of false and true predictions (Fig. 8a). Nevertheless, IIS did predict key inflammatory markers, including TXB2 and PGE2 as reflected by AUC values close to 1, signifying a majority of accurate predictions (Fig. 8a). We then leveraged data encompassing IIS to perform PCA to identify potential similarities among RDEB adults (Fig. 8b). The analysis clustered RDEB adults into two distinct groups (p-value: 0.005), moderate/high and highest IIS. To pinpoint factors that may allow the differentiation between these two clusters, we conducted a comparative analysis of key cytokine and lipid levels within each group. This analysis revealed a substantial increase in pro-inflammatory cytokines (IFNγ, IL17a, IL6, TNF) and anti-inflammatory cytokine (IL10), along with pro-inflammatory lipid mediators TXB2, PGE2, and LTB4 in patients from cluster 2 (TC062, BA525, YZ319, GM990) who exhibited high IIS. Despite the limited size of the RDEB cohort, these predictions underscore the importance of immunometabolic/inflammatory aspects toward more personalized standard care of RDEB adults.

## Discussion

Our immunometabolic profiling accentuated the immune-inflammatory dimension in RDEB adults with active/effector T and dysfunctional NK cells, concomitant with an overall pro-inflammatory lipid signature. While awaiting full clinical translation of COL7A1 genetic corrections, our artificial intelligence interrogation, which emphasized the phenotype-endotype association in RDEB adults, suggests that monitoring of immune and inflammatory metabolic states could improve the management of these patients.

Besides the anomalies in innate immunity, impairment in adaptive immunity in RDEB has been suggested but remained elusive[2,13]. Our in-depth exploration of T cells revealed an overabundance of effector cell subsets including terminally differentiated ones, alongside the emergence of a particular activated/inflammatory signature, whether related to genetic or environmental factors. A time-dependent increase in adaptive T cell infiltrate with a shift from activation to that of exhaustion occurs in RDEB mice skin and correlates with progressive fibrosis in these mice[11,37]. Similar findings were observed in human RDEB skin from different stages of fibrosis/disease[11,37]. The IMC analysis showing remarkable high levels of activated CD4+ and CD8+ T cell infiltration co-localising with fibrotic region in skin lesions of RDEB adults, further support these findings and underscore the interrelation between systemic immunity and disease pathophysiology.

Indeed, altered proportion of effector and memory T cell subsets is a common feature of chronic inflammatory and infectious diseases and can be attributed to elevated levels of inflammatory cytokines[38,39]. A similar scenario might occur in RDEB adults displaying high levels of various pro-inflammatory cytokines, including IL6, IFNγ, and IL17a, in the periphery. Interestingly, the highly plastic CCR6+CD161+ CM Th1 cells, also named non-classical Th1 endowed with high capacity to produce pro-inflammatory IFNγ and IL17a[40], were enriched in RDEB patients.

The above T cells' signature of RDEB is in line with their single-cell immunometabolics showing altered metabolic contexture marked by increased glycolytic and FAAO capacity as well as higher levels of protein synthesis. The functional plasticity of immune cells is often shaped by the local inflammatory environment characterized by drastic changes in nutrient and/or metabolite abundance, metabolic

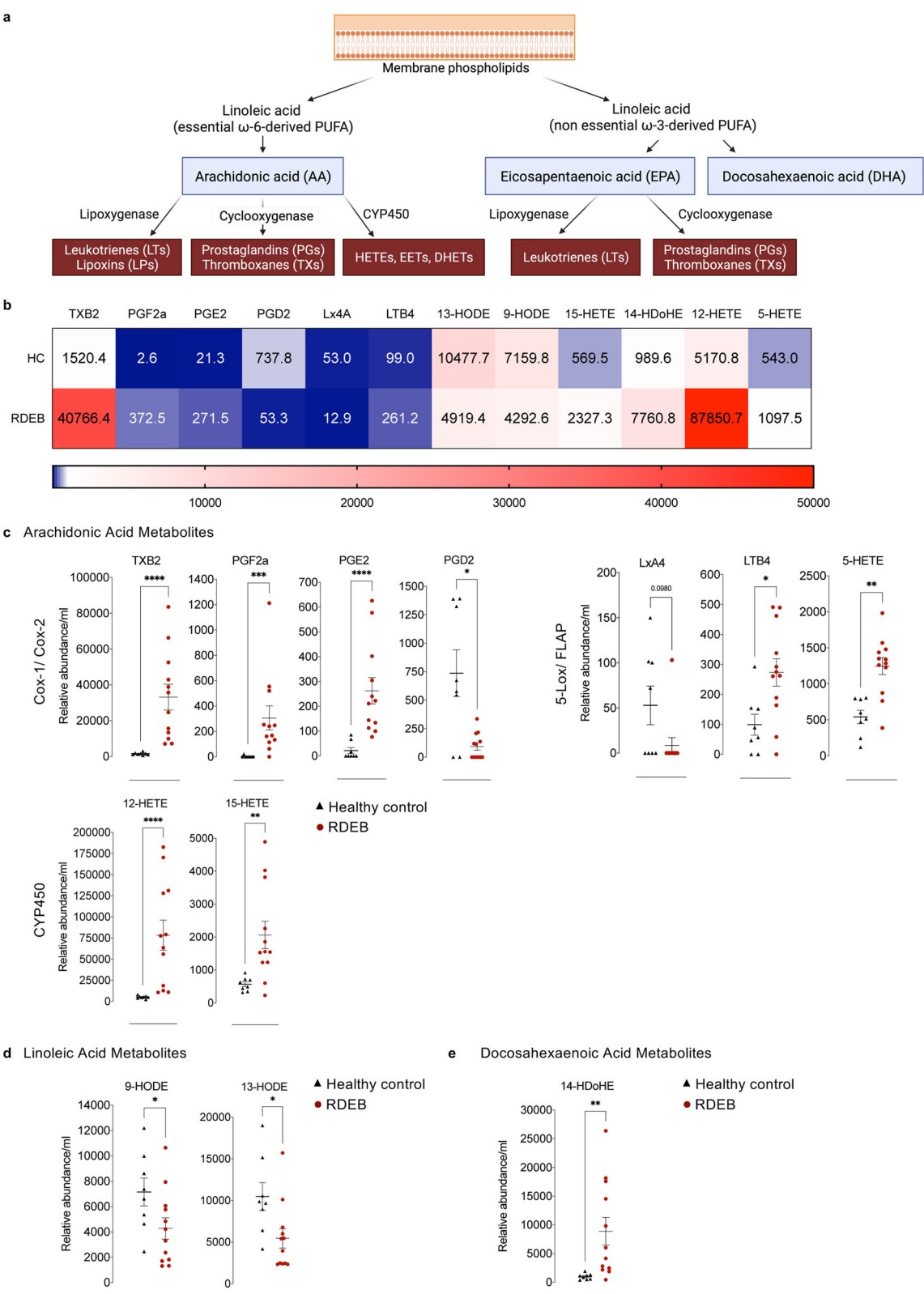

**Fig. 6 | RDEB adults nurse a particular eicosanoids signature. a** Schematic representation of lipid mediators downstream polyunsaturated fatty acid (PUFA). **b** Heatmap representing mean values of serum eicosanoid lipid mediators' relative abundance from healthy controls ($n = 8$) and RDEB adults ($n = 12$). Colours vary according to lipid abundance density. **c–e** Relative abundance of altered eicosanoids derived from acid arachidonic (AA) downstream cyclooxygenase (COX), lipoxygenase (LOX/FLAP), and cytochrome P450 (CYP450) pathways, linoleic acid (LA), and docosahexaenoic acid (DHA). Lipid mediators are presented in scattered plots as mean values of relative abundance/ml of sera ± SEM from 8 healthy controls and 12 RDEB adults. Healthy controls (black triangles) and RDEB adults (red circles). Statistical analysis is performed with two-sided unpaired *t*-test. Asterisks represent the significant differences between healthy controls and RDEB adults (*$p < 0.05$, **$p < 0.01$, ***$p < 0.001$, ****$p < 0.0001$). Source data are provided as a Source Data file.

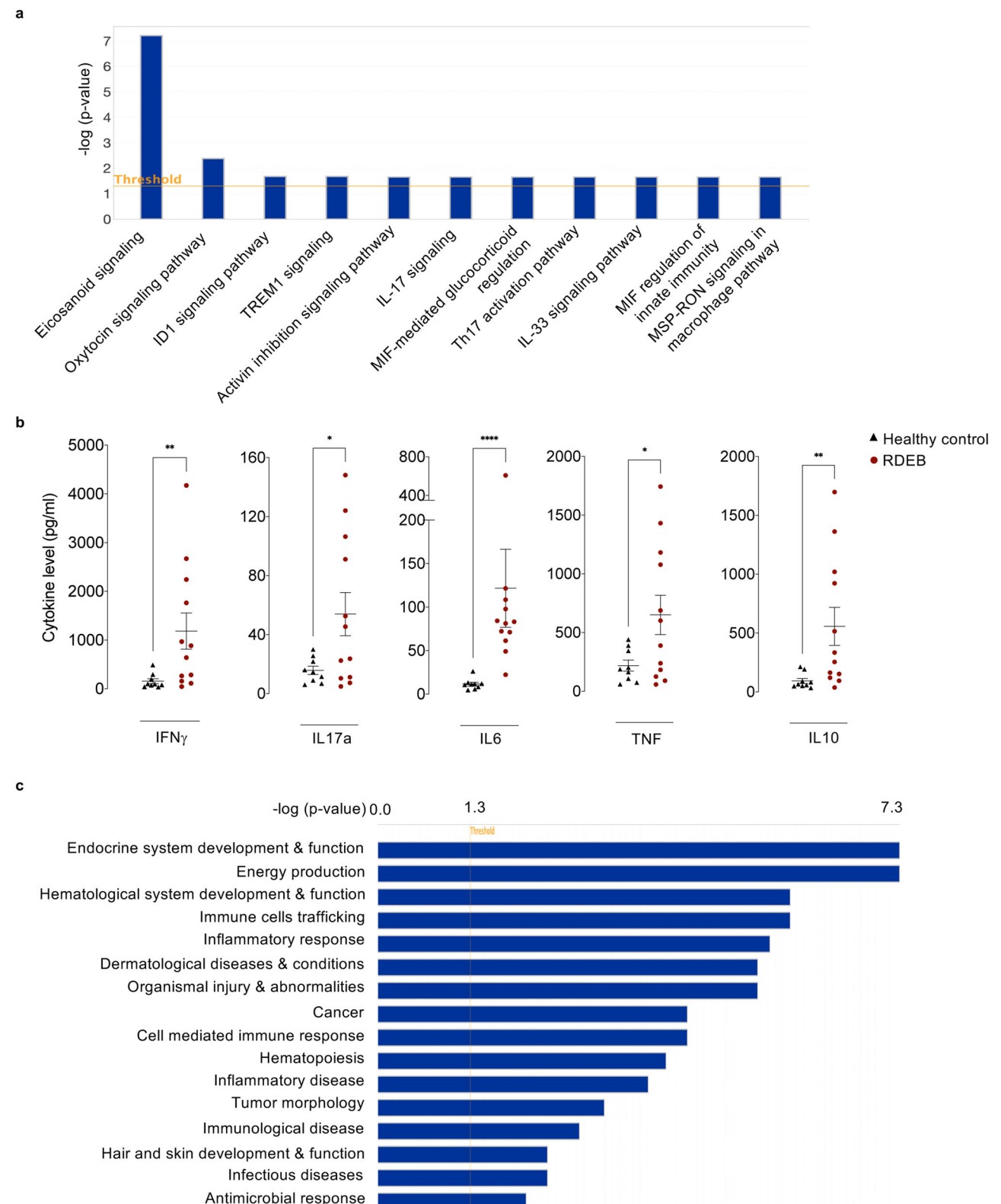

**Fig. 7 | Mapping lipid rewiring in RDEB. a** Ingenuity pathway analysis (IPA) predicted canonical pathways upregulated and potentially involved in RDEB pathophysiology. The Y-axis shows the -log (p-value). The orange line indicates a threshold at -log (p-value) = 1.3, which represent p-value = 0.05. Statistical analysis was performed with two-sided fisher exact test. **b** Cytokines level (pg/ml) in serum samples presented in scattered plots as mean values of relative abundance/ml of sera ± SEM from 9 healthy controls and 12 RDEB adults. Statistical analyses were performed with two-sided unpaired t-test. Asterisks represent the significant differences (*p < 0.05, **p < 0.01, ****p < 0.0001). Source data are provided as a Source Data file. **c** IPA predicted diseases and bio-functions possibly related to RDEB. The x-axis shows the -log(p-value). The threshold at -log (p-value) = 1.3 is indicated. Statistical analysis was performed with two-sided fisher exact test.

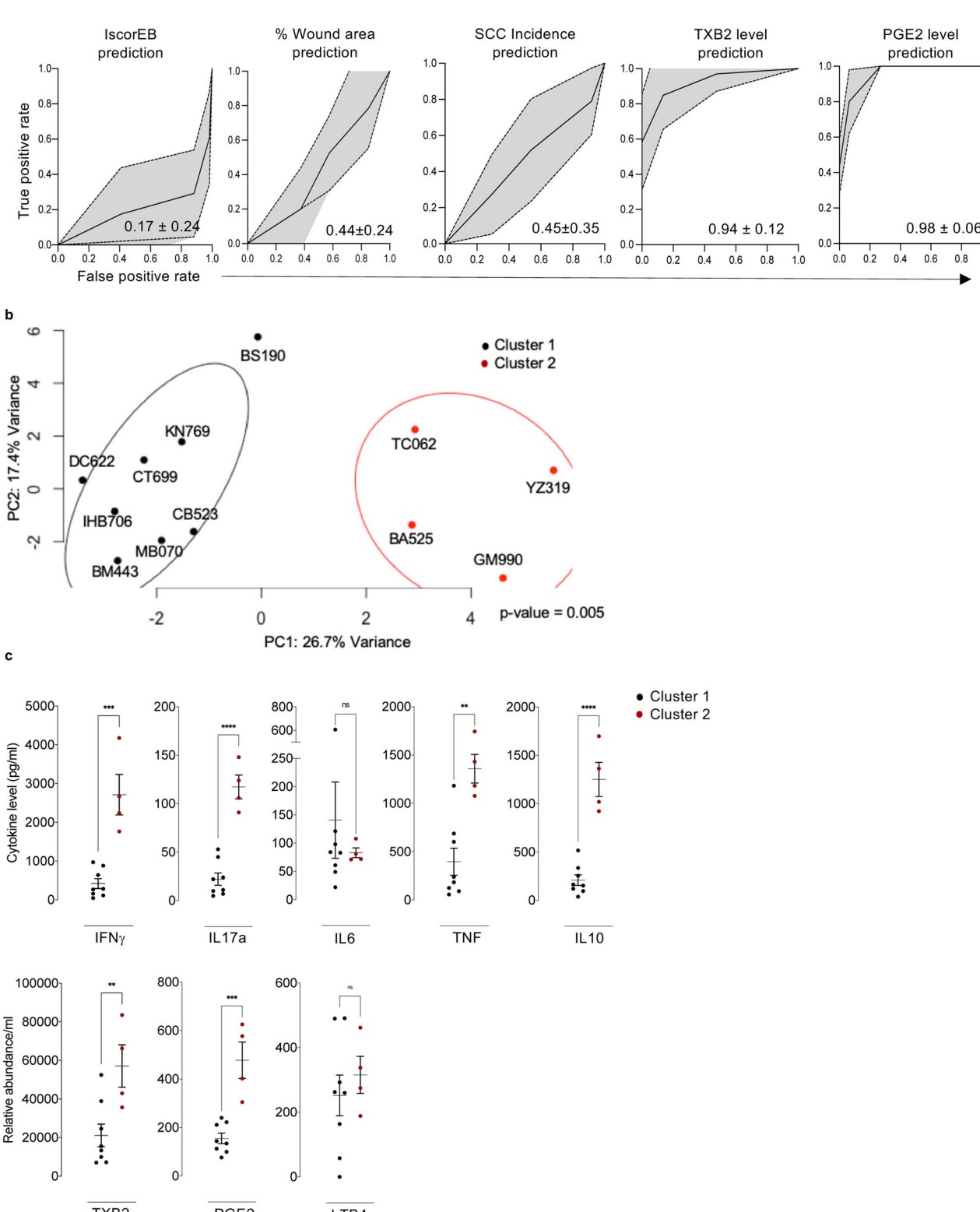

**Fig. 8 | RDEB beyond skin and mucosa. a** Receiver operating characteristic (ROC) curves prediction of IscorEB, % wound area, SCC incidence, TXB2 and PGE2 level in RDEB patients. Prediction is based on Inflammation Immunity Score (IIS) data. The *Y*-axis shows the true prediction, while the *X*-axis shows false prediction. Curves are mean ± SD of curves coordinates obtained following 50 prediction iterations with independent training/validation cohort sampling. Mean area under the curve ± SD are indicated on graph. **b** PCA of RDEB adults IIS. Patients are distributed in 2 significant clusters. Clustering significance was determined using Permutational multivariate analysis of variance (PERMANOVA), and *p*-value is indicated on the graph. **c** Cytokines levels (pg/ml) in RDEB adults sera based on PCA clustering (upper panel). Relative abundance of eicosanoids (lower panel), presented as relative abundance/ml of sera. Results are presented in scattered plots as mean values ± SEM from 8 RDEB adults (cluster 1) and 4 RDEB adults (cluster 2). Statistical analyses were performed with two-sided unpaired *t*-test. Asterisks represent the significant differences (**$p < 0.01$, ***$p < 0.001$, ****$p < 0.0001$). Source data are provided as a Source Data file.

acidosis, hypoxia, and local stimuli including cytokines. The role of pro-inflammatory signalling in promoting glycolysis through the expression of glycolytic enzymes and the activation of glycolytic pathways has been reported[41]. For instance, the expression of CPT1A, a rate-limiting enzyme of mitochondrial fatty acid oxidation, is induced by the presence of IL17a[42]. The high plasticity of RDEB adults' T cells and their ability to adapt their metabolism to the local environment enriched in IFNγ, TNF and IL17a, among others, is highlighted by the observed decrease in glucose and mitochondrial dependence. While RDEB T cells showed a decreased mitochondrial dependency, the upregulation of their OXPHOS-dependent FAAO capacity, witnesses proper mitochondrial function. Of note, the SCENITH approach cannot distinguish whether the latter is related to increased FA and/or AA oxidation capacity.

NK cell dysfunctionality in RDEB patients is a notable finding of our single-cell energy metabolic profiling combined to functional analysis. Exhaustion, anergy, and senescence can all lead to NK cell dysfunction[26]. In agreement with pioneer investigations[21], RDEB NK cells showed major defects in their cytotoxic and cytokine-release functions probably linked to the expression of PD-1, LAG-3, and NKG2A markers that are often associated with impaired NK cell activity[43,44]. Furthermore, these cells displayed increased NKG2C and KIRD2 receptor expression concomitant with decreased NKG2D expression. While defining "memory NK cells", such a profile has been also associated with the emergence of CD56$^{dim}$CD57$^{+}$ senescence-related phenotype, which we found abundant in RDEB adults. The metabolic fingerprints of RDEB NK cells, although nurtured variability that hampered reaching significance, point to an exhausted/senescent cell fate resembling NK cells in cancer and chronic infection microenvironment[45]. Our IMC analysis did not show NK cells within the immune cell infiltrate of skin lesions from RDEB adults. Whether this is related to the exhausted/senescent-related phenotype of RDEB adults peripheral NK cells that distresses their homing to skin injuries cannot be affirmed but might be conceivable. Without ruling out the potential impact of genetic elements, the observed NK cell dysfunctionality may thus underlie the high incidence of infections and tumours in RDEB adults. Together, the observed metabolism-dependent disruption of immune cell phenotype and function is illustrated in both T and NK cell compartments of RDEB adults. Consequently, the metabolic alterations in RDEB that imprint various immune components, would create a conducive environment favouring chronic wounding, infections and cancer.

Besides cytokines, inflammatory processes are intrinsically related to bioactive lipid mediators which act as wildfire-starters of inflammatory pathways. Decades ago, AA and LA were suggested as potential factors contributing to RDEB[46]. The diversity of RDEB adults' lipid signature marked by the accumulation of AA-derived pro-inflammatory metabolites concomitant with a considerable decrease of anti-inflammatory mediators, recapitulates their persistent inflammatory status. AA and its downstream metabolites produced by innate cells including neutrophils and monocytes[47,48] have been implicated in the pathogenesis of various disorders[49] including skin inflammation with oedema that are indistinguishable from RDEB skin blisters[46]. Given that our RDEB cohort was free from opportunistic infections at the time of sampling, the elevated levels of pro-inflammatory metabolites concord with the observed excessive neutrophils counts. While we cannot rule out the possibility of prior infections contributing to this, the increased neutrophil number in our cohort might also be related to a continuous surge from bone marrow into the circulation, similar to what is observed in other diseases[50,51]. This lipid inflammatory signature is also in line with the shift of CD14$^{+}$CD16$^{-}$ classical and CD14$^{-}$CD16$^{++}$ non-classical monocytes toward the pro-inflammatory CD14$^{+}$CD16$^{+}$ intermediate monocytes, a major source of cytokines, and are associated with various diseases[52,53]. Activated/inflammatory monocytes/macrophages expressing major histocompatibility complex (MHC) class II molecules are major immune infiltrate in RDEB mice and human skin lesions[37], and our IMC analysis showed elevated numbers HLA-DR-positive monocytes/DC/macrophages in skin lesions of RDEB adults. The domination of inflammatory CD14$^{+}$CD16$^{+}$ intermediate monocytes subset in the periphery of RDEB adults over other monocyte subsets is in line with these findings, although a direct link cannot be yet established. Nonetheless, our findings may link persistent inflammation in RDEB adults to a dysregulated AA and LA metabolism and suggest that neutrophils and monocytes might nurture the RDEB inflammatory landscape and unhealed chronic wounding.

Immunological changes observed in adult patients can result from decades of extensive skin wounds and recurrent infections. These factors could lead to a hyperimmune response aimed at protecting RDEB patients from environmental insults and severe infections in the absence of an intact epidermis. Our observed active/effector T cell signature, along with spatial analysis of skin wounds, aligns with this understanding. However, wound-related hyperimmune activation might not be the sole explanation. Interestingly, RDEB adults who exhibited the highest immune inflammation signature did not consistently have the most extensive wounds. This suggests that intrinsic dysregulation of systemic immunity may also be a contributing factor, its involvement in adaptive immunity remains to be demonstrated. While a role for Coll-VII in regulating innate immunity has been put forward[7], its potential involvement in adaptive immunity has yet to be explored. Nonetheless, the presence of Coll-VII in the thymus basement membrane and secondary lymphoid organs[7,13,54] suggests a possible role in regulating adaptive immune cells. Mutations in *COL7A1* may contribute to intrinsic dysregulation of immune homoeostasis in RDEB patients, leading to systemic immune dysregulation and chronic inflammation, which can result in fibrosis and may be linked to the development of squamous cell carcinoma (SCC). Our in-depth analysis, supported by artificial intelligence prediction modelling, advocates the systemic challenges associated with RDEB, which extend far beyond issues with the skin and the mucosa. Our PCA encompassing IscorEB, wound area, SCC history, and IIS further supports this notion, categorizing patients into two main groups: moderate/high and highest immune/inflammatory profiles. The convergence of multiple factors, including an inflammatory microenvironment, creates conditions that may promote the development of SCC, a significant cause of morbidity and mortality in RDEB patients[35]. Among those patients with the highest inflammatory profile, only half developed SCC. Although a direct correlation between inflammation/immunity and SCC cannot be drawn, all patients in this group displayed high levels of the inflammatory cytokine IL6, a known risk factor for SCC[55]. Hence, we cannot rule the possibility that SCC-free patients at recruitment may still develop this type of cancer. The clinico-pathophysiology of RDEB is highly complex, with genetic variants being the only well-established factor causing blistering and wounds. However, it remains unclear whether other factors contribute to disease progression. Our study investigates the potential role of peripheral immune, inflammatory, and metabolic factors in adult RDEB patients. We found that the IIS may represent an additional parameter in understanding this disease. Our prediction model underscores the importance of incorporating the immune-inflammatory status as an assessment parameter of disease to refine the IscorEB, ultimately leading to improved patient management.

RDEB is a very rare disease with an estimated prevalence of one patient in two million. Patients included in our study were diagnosed with *COL7A1* variants that led to gene splicing, frameshift, or introduction of early stop codons. Thus, while providing an in-depth definition of the immunometabolism inflammatory signature of RDEB adults, our study is limited by the small size of our cohort, the limited number of skin biopsies, and the challenges in obtaining sufficient quantities of various immune cell subsets from these suffering patients for further functional mechanistic studies. Despite these constraints,

our findings—at the intersection of metabolic signalling, immune cell fate, and system immunology—underscore the concept of RDEB as genetic disorder with distressed immunometabolism/inflammation. In addition, they emphasize the need to investigate larger retrospective paediatric and adult cohorts, as highlighted in previous studies[33,56], to fully understand the disease and refine this concept. Nonetheless, our findings could be valuable for both clinicians and researchers, potentially laying the groundwork for a more tailored personalized monitoring and classification of RDEB beyond the skin injuries.

The immune system is a highly strategic and plastic network that orchestrates both health and disease. Reprogramming the immune inflammatory status has emerged as a promising and attractive promising management strategy for various conditions, including RDEB[15–17,22,57–62]. Without ruling out the importance of gene editing to correct Coll-VII deficiency[63,64], our study highlights the phenotype-endotype associations in human RDEB and emphasizes the importance of including the suggested IIS as an additional assessment parameter to refine the IscorEB. Our findings underscore the necessity for further development of systemic strategies aimed at fine-tuning peripheral immune cell homoeostasis to lessen inflammation without compromising immune defence. Ultimately, this approach could lead to better management and improved quality of life for long-suffering RDEB adults.

## Methods

### Study design and participants

To elucidate the immunometabolic/inflammatory landscape of the human incurable RDEB, we used fresh blood samples from a small clinical-biological cohort of adult patients (8 females, 4 males) and skin biopsies from two patients (Supplementary Table 1 and 2), followed and managed at the Dermatology Department—French national reference center for rare diseases of the skin and mucous membranes of genetic origin (MAGEC)—Saint-Louis Hospital, in comparison to healthy donors. All participants signed informed consent following human ethics committee "Comité consultatif pour la protection des personnes dans les recherches biomédicales" (Direction de la Recherche Clinique, de l'Innovation des relations avec les universités et les organismes de recherche (DRCI), AP-HP, Saint-Louis Hospital). Patients were sampled (blood and skin biopsies) outside of any clinical opportunistic infection or active cancer. All protocols were approved by the institution (CPP No. 2023-A01351-44) and were conducted in accordance with guidelines from the Declaration of Helsinki. As readouts, we applied a combination of single-cell mass cytometry (CyTOF)[65], imaging mass cytometry (Hyperion Imaging System)[66] and energetic metabolism by profiling translation inhibition (SCENITH)[67], as well as functional assays to monitor immune cells, and subjected serum to large-scale quantitative shotgun lipidomic coupled to mass spectrometry (LC-MS/MS) and cytokine profiling to screen lipids and cytokines, respectively. All peripheral readouts were obtained with the same blood sample for each patient or healthy donor. Skin biopsies readouts obtained with skin biopsies from RDEB patients were compared to healthy tissue sections obtained from patients who underwent breast reduction plastic surgery. To integrate our findings, we made benefits from artificial intelligence and machine learning.

### Mass cytometry—cytometry by Time-Of-Flight (CyTOF), flow cytometry

Fresh whole blood samples were collected in sodium heparinized tubes and immune cell subsets were determined by CyTOF XT Mass Cytometer. According to the manufacturer protocol, we start with an identical number of total immune circulating cells to perform the acquisition. A fixed number of 500,000 CD45-positive circulating immune cells were acquired for each patient and control to avoid any eventual bias in the analyses. The gating strategy is depicted in Supplementary Fig. 1a. Briefly, whole blood samples were stained by adding 270 μl of blood directly to a Maxpar Direct Immune Profiling Assay (#201334) combined with the Maxpar Direct Myeloid and B cell Expansion Panel 1 (#201402) (Standard Bio Tools, San Francisco, CA, USA). The panel of antibodies is listed in Supplementary Table 3. Samples were mixed and incubated for 30 min at room temperature. Red blood cells were lysed using ACK lysing buffer (Gibco A1049201, Thermo Fisher Scientific, USA). Samples were then washed with Cell Staining Buffer, fixed with 1.6% Paraformaldehyde for 15 min at room temperature, permeabilized using Fix Perm buffer, then incubated with 125 nM Iridium DNA intercalator (Standard Bio Tools) overnight. Cells were washed and resuspended at a concentration of $10^6$ cells/mL in Maxpar Cell Acquisition Solution and mixed with 10% of EQ Beads immediately before acquisition. Samples were acquired on CyTOF XT at an event rate ≤500 events/s. Dual count calibration, noise reduction, cell length threshold between 10 and 150 pushes, and a lower convolution threshold equal to 10 were applied during acquisition. Mass cytometry standard files produced by the CyTOF-XT were normalized using the CyTOF software version 6.7.1014 (Standard Bio Tools). Unsupervised clustering was performed on the expression values of the markers using the FlowSOM or Phenograph algorithm (OMIQ, www.omiq.ai). The median expression level of markers across all clusters were visualized using heatmap. We then applied the nonlinear dimensionality reduction technique, Uniform Manifold Approximation and Projection (UMAP) or Automated Optimized Parameters for T-distributed Stochastic Neighbor Embedding (opt-SNE). Frequency (%) of different immune cells subsets is obtained through reduction and clustering based on various cell surface markers. Absolute counts were determined using the following calculation: Absolute count of an immune cell subset = white blood cells count (from each RDEB patient, or standard average count of healthy individuals) x 1000 x percent of the immune cell subset.

The expression of NKG2D, KIR2D, PD-1, LAG-3, NKG2A, NKG2C on NK cells was determined by staining with specific antibodies (Supplementary Table 7) and flow cytometry. Cells were acquired on BD Fortessa flow cytometer and then analysed using FlowJo software (10.8.1).

### Imaging Mass Cytometry (IMC) by hyperion imaging system

Formalin-Fixed Paraffin-Embedded (FFPE) sections of 4 μm thickness from skin, were cut onto glass slides. Sections were de-paraffinized with xylene and carried through sequential rehydration from 100% Ethanol to 70% Ethanol before being transferred to Tris-buffered saline (TBS). Heat-induced antigen retrieval was performed in a water bath at 95 °C for 30 min in Tris/EDTA buffer (10 mM Tris, 1 mM EDTA, pH9). Slides were cooled to room temperature (RT) and were subsequently blocked with Phosphate-buffered saline (PBS)-3% BSA for 30 min at room temperature (RT). Each slide was incubated with 100 μl of the antibody cocktail (Supplementary Table 4) overnight at 4 °C. Then, slides were washed 3 times with PBS and labelled with 1:500 dilution of Intercalator-Ir (Standard BioTools) in TBS for 2 min at RT. Slides were briefly washed with H2O and air dried the acquisition was performed with Hyperion Imaging System (Standard Bio Tools). Data were acquired on a Hyperion imaging system coupled to a Helios Mass Cytometer (Standard BioTools), at a laser frequency of 200 Hz and laser power of 3 dB. For each recorded ROI, stacks of 16-bit single-channel TIFF files were exported from MCD binary files using MCD Viewer 1.0 (Standard Bio Tools). Cell-based morphological segmentation was carried out by using YOUPI software[68]. To extract quantitative data, FCS files were uploaded and analysed with OMIQ software. Single-cell expression data were Arcsinh transformed with 1 as a cofactor before analysis. Markers used for clustering were limited to the most informative in distinguishing immune cell populations and those deemed to have an acceptable signal-to-noise profile: CD14, CD204, DC Lamp, CD16, CD163, Foxp3, CD4, DCsign, CD68, CD20, CD8, CD138, MPO, CD3, CD206, HLA DR and CD15. We used the average protein expression profile for each cluster to determine cell

lineage on the basis of previous markers. Where average expression profiles were similar with respect to these markers between several clusters, these cluster were merged. To visualize the high-dimensional data in two dimensions, the opt-SNE algorithm was applied on data using the previous markers.

## T cell proliferation assay

Peripheral blood mononuclear cells (PBMC) were prepared from blood samples of healthy controls and RDEB patients by centrifugation on a Ficoll-Hypaque density gradient. T cell proliferation was then determined as we described[57]. Briefly, cells were labelled with Carboxyfluorescein succinimidyl ester (CFSE) (2.5uM) (C34554, Thermo Fisher, USA), then $10^5$ CFSE-labelled PBMCs were stimulated with Immunocult CD3/CD28/CD2 antibody complexes at a concentration of 1.5 µL/$10^5$ cells in 200 µL (10971, StemCell, Grenoble, France) for 4 days. Cells were then harvested, labelled with CD3-BUV805, CD4-APC-H7, CD8-pacific-blue specific antibodies (Supplementary Table 7) and proliferation of CD4$^+$ and CD8$^+$ T cells was determined as loss of CFSE fluorescence using flow cytometry. Cells were acquired on BD Fortessa flow cytometer and then analysed using FlowJo software (10.8.1).

## NK cells cytotoxicity and degranulation assay

NK cells cytotoxicity against K562 target cells labelled with CFSE (2.5 µM) was assessed as we described[69]. Briefly, target cells were cultured alone or in the presence of IL-15-primed RDEB or healthy controls PBMC at an effector to target ratio of 10:1 with PeCy7-conjugated anti-CD107a mAb (as a marker of NK cells degranulation) for a total of 4 hrs. BD GolgiPlug/Stop protein Transport Inhibitors (554724 and 555029, BD Bioscience) were added, cells were harvested, labelled with CD3-BUV805 and CD56-APC specific antibodies (Supplementary Table 7) and stained with 7-aminoactinomycin D (7-AAD) (BD biosciences, le Pont-de-Claix, France). The expression of CD107a was analysed on CD3$^-$CD56$^+$ NK cells and the percentage of 7-AAD-positive cells (dead cells) in CFSE-positive target cells was then determined by flow cytometry. Cells were acquired on BD Fortessa flow cytometer and then analysed using FlowJo software (10.8.1) Specific lysis was calculated according to: 100 x [(observed lysis-spontaneous lysis)/(100-spontaneous lysis)].

## Cytokines assays

Cytokine levels in healthy control and patients' sera, collected using BD vacutainer tubes following the standard operating procedure (SPO), were evaluated using the flow cytometry bead-based array with a MACSPlex Mix Cytotoxic basic kit (130-125-767) according to the manufacturer's procedures (Miltenyi Biotec, Bergisch Gladbach, Germany). MACSPlex cytokine kit detects human IFNγ, IL6, IL10, IL17A and TNF. Samples were acquired using a Canto II (BD) and analysed using FlowJo (10.8.1) software.

IFNγ and TNF produced by NK cells were evaluated by intracellular staining. Briefly, $10^6$ PBMC were stimulated with PMA (50 ng/ml)/ionomycin (1 µg/ml) for 4 hrs in the presence of BD GolgiPlug/Stop protein Transport Inhibitors, then labelled with anti-CD3, anti-CD56, anti-IFNγ and anti-TNF (Supplementary Table 7) using BD Cytofix/Cytoperm Fixation/Permeabilization Kit (554714). Cells were acquired on BD Fortessa flow cytometer and then analysed using FlowJo (10.8.1) software.

## Lipid extraction and quantitative analysis by LC-MS/MS

Sera from either healthy controls or RDEB patients were snap-frozen in liquid nitrogen and stored until lipid extraction and analysis as we described[70]. Briefly, 50 µL of sera were used for protein quantification using Bio-Rad Protein assay. Internal standards (IS) for neutral lipids and deuterated IS for bioactive lipids were purchased from Sigma-Aldrich, MO, USA and Cayman Chemicals, MI, USA, respectively. For neutral lipids profiling, equivalent 50 µL of sera were extracted according to Bligh and Dyer method in dichloromethane/methanol/

water (2.5/2.5/2.1: v/v/v), in the presence of IS. The organic phase was evaporated to dryness and neutral lipids were separated through a Solid Phase Extraction (SPE) cartridge. 1 µL of the lipid extract was analysed by gas-liquid chromatography on a FOCUS Thermo Electron system using Zebron-1 Phenomenex fused silica capillary columns (5 m X 0,32 mm i.d, 0.50 m film thickness). Relative quantification was performed and values are expressed as (metabolite area/ISTD area)/mL. For free Oxylipins/Bioactive lipids, equivalent 50 µL of serum sample was used to extract free oxylipins. In brief, 300 µL of cold methanol and 5 µL of internal standards (Deuterium labelled compounds of LxA4-d5, LtB4-d4, 5-HETE-d8 at the concentration of 400 ng/mL) were added to each sample, centrifuged, submitted to SPE. Lipids mediators were then eluted then subjected to LC-MS/MS analysis. Data were acquired in MRM mode with optimized conditions (ion optics and collision energy). Peak detection, integration, and quantitative analysis were performed using Mass 19 Hunter Quantitative analysis software (Agilent Technologies).

## SCENITH

Cryopreserved peripheral blood mononuclear cells (PBMC) were thawed and plated in 96-well plates ($5.10^6$ cells/mL). Briefly, cells were treated for 15 min with DMSO (Sigma-Aldrich), 2-Deoxy-D-Glucose (100 mM, Sigma-Aldrich), Oligomycin (1µM, Sigma-Aldrich) or a sequential combination of both drugs. Puromycin (10 µg/ml, Sigma-Aldrich) was added for 20 min at 37 °C. Cells were washed in cold PBS and stained with Live/Dead Fixable Blue Dead Cell Stain Kit (Thermo-Fisher Scientific). Non-specific binding was blocked by the human TruStain FcX (Biolegend). Surface staining was carried out in BD Pharmingen stain buffer (BD Biosciences). Surface antibodies used in SCENITH experiments are listed in Supplementary Table 7. For intracellular staining, cells were fixed and permeabilized using fixation/permeabilization kit for FOXP3 Transcription factor staining (Ebioscience) following manufacturer's instructions. Monoclonal anti-Puromycin Alexa Fluor 647 conjugated antibody (Clone 12D10, Merckmillipore) was used for intracellular staining of puromycin. Cells were analysed on a Symphony A5 cytometer (BD Biosciences). Glucose dependence was calculated as: 100 × (puromycin MFI levels of DMSO-treated cells−puromycin MFI levels of 2-DG-treated cells)/(puromycin MFI levels of DMSO treated cells−puromycin MFI levels of 2-DG + oligomycin treated cells). Mitochondrial dependence was calculated as: 100 × (puromycin MFI levels of DMSO-treated cells−puromycin MFI levels of oligomycin-treated cells)/(puromycin MFI levels of DMSO-treated cells−puromycin MFI levels of 2-DG + oligomycin-treated cells). Glycolytic capacity was calculated as: 100% mitochondrial dependence. FAAO capacity was calculated as: 100% Glucose dependence.

## ROC prediction based on Inflammation Immunity Score (IIS)

The association between IIS and clinical IscorEB, % wound area, SCC incidence, TXB2, and PGE2 levels was assessed through the ability to predict the later based on IIS in RDEB patients. Prediction of clinical IscorEB (outcome ABOVE or BELOW based on median), TXB2, and PGE2 levels (outcome for each: ABOVE or BELOW based on median) was based on IIS dataset containing abundance values of cytokines and lipids, as well as absolute count for immune cell populations. Receiver operating characteristic (ROC) curves were calculated (R version 4.1.2, randomForest 4.7-1.1 package, ROCR package) using training data set and validation data set containing randomly affected 70% and 30% of RDEB patients, respectively. ROC calculation was repeated 50 times with random sampling of the training and validation data. Area under curve (AUC) is measured for each iteration. Mean AUC and standard deviation are presented for each graph.

## Quantification and statistical analysis

Statistical analysis was performed using GraphPad Prism 9.5.0 (GraphPad Software, Inc., San Diego, CA, USA). Data on the graphs

represent the mean ± SEM from each group and comparisons were performed using unpaired *t*-test assuming Gaussian distribution with parametric test if normality is obtained with Anderson-Darling (A2*)/ D'Agostino-Pearson omnibus (K2)/Shapiro-Wilk (W)/Kolmogorov-Smirnov (distance), or non-parametric test with Mann Whitney test if normality is not obtained. *P*-values ≤0.05 were considered statistically significant. Based on their cell-surface markers, clusters obtained from PBMCs were subjected to Component Analysis (PCA) using R (R version 4.1.2, package pca2d). Clustering significance was determined using Permutational multivariate analysis of variance (PERMANOVA). Correlation matrix was computed using R (R version 4.1.2, package corrplot 0.92).

## Reporting summary

Further information on research design is available in the Nature Portfolio Reporting Summary linked to this article.

## Data availability

The CyTOF, and SCENITH datasets generated and analysed during the current study have been deposited in the FlowRepository (http://flowrepository.org) under accession code and FR-FCM-Z8NM, respectively. All other data are available in the article and its Supplementary files or from the corresponding author upon request. Source data are provided with this paper.

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

## Acknowledgements

The work was supported by AO MARS 2020-la Société Française de Dermatologie, AP-RM-22-006-Fondation d'Avenir, MEDIMACS-ANR-17-HDIM-0005-02, MetaboHUB-ANR-11-INBS-0010 and funding provided by Inserm, CNRS, and the University of Toulouse for Jabrane-Ferrat, as well as Inserm funds for UMRS976. The authors thank Pr. D. Charron for critical reviewing of the manuscript, Niclas Setterblad and Christelle Doliger from Technology platform of Saint-Louis Research Institute (Paris University, Paris, France), CyPS—Pitié-Salpêtrière Mass Cytometry (CyTOP) Platform (Sorbonne University, Paris, France), Justine Bertrand and Pauline Lefaouder from the MetaToul-Lipidomic platform (I2MC, Inserm, Toulouse, France), L. Lobjoie and S. Allart from (Infinity Cell Imaging core facility), Snaigune Miskinyte (COL7A1 genetic testing, INSERM UMR1163), and Y. Delarue from (Cytometry core facility Hyperion, Brest, France).

## Author contributions

R.D. conceived the study. R.D., N.H., E.M., S.S-D., H.E.C. and N.J.F. designed experiments. N.H., E.M. and H.E.C. performed experiments.

A.C., P.H. performed mass cytometry. R.D., N.H., E.M, Q.C., H.E.C., H.L. and N.J.F. analysed the data. C.D. performed AI studies. J-D.B., P.B., E.B. and A.H. clinicians supporting the study and critical review. A.H. supervised genetic analysis of COL7A1. R.D., and N.J.F. acquired funding and supervised experiments and data analysis. R.D., N.H. and E.M. wrote the manuscript. N.H. and E.M. contributed equally to this work and share first-authorship, the first was determined alphabetically. F.A., N.J.F., and R.D. critically reviewed and approved the manuscript.

## Competing interests

The authors declare no competing interests.

## Additional information

[1]National Institute of Health and Medical Research (INSERM) UMRS-976 HIPI, Paris Cité University, Saint-Louis Hospital, 75010 Paris, France. [2]Boston Childrens Hospital, Harvard Medical School, Boston, MA 02115, USA. [3]INSERM U1016, The National Centre for Scientific Research (CNRS) UMR 8104, Paris Cité University, 75014 Paris, France. [4]Pitié-Salpêtrière Cytometry, UMS037, Sorbonne University, 75013 Paris, France. [5]LBAI, INSERM UMR1227, Brest University, 29200 Brest, France. [6]Genethon, 91000 Every-Courouronnes, France. [7]Dermatology Department, AP-HP, Saint-Louis Hospital, 75010 Paris, France. [8]Laboratory of Genetic Skin Diseases, Imagine Institute, Paris Cité University, INSERM UMR 1163, 75015 Paris, France. [9]Division of Immune and Infectious Diseases, CHU de Quebec Research Centre, Department of Microbiology-Infectiology and Immunology, Faculty of Medicine, Laval University, Quebec City, QC, Canada. [10]Institute for Infectious and Inflammatory Diseases, CNRS UMR5051, INSERM UMR1291, Toulouse III University, 31059 Toulouse, France. [11]These authors contributed equally: Nell Hirt, Enzo Manchon. [12]These authors jointly supervised this work: Nabila Jabrane-Ferrat, Reem Al-Daccak.
✉e-mail: reem.al-daccak@inserm.fr

