## [Transparent Peer Review file · Nature Communications]

Systems immunology integrates the complex endotypes of recessive dystrophic epidermolysis bullosa

Corresponding Author: Dr Reem Al-Daccak

Version 0:

Reviewer comments:

Reviewer #1

(Remarks to the Author)

Comments to the authors:

This paper is a timely and comprehensive study that applies systems immunology approaches utilizing single-cell high-dimensional techniques to capture the immune cell signatures and metabolic profiles diversity in RDEB adult patients. The patients were sampled outside of any opportunistic infection or active cancer. The study reveals the distinctive inflammation and immunity characteristics of RDEB patients, with signatures of active/effector T cells and dysfunctional natural killer cells, along with an overall pro-inflammatory lipid signature—a novel finding in the field of RDEB.

The authors suggest, through primarily computer-based analyses, that RDEB is not solely a cutaneous disease but rather exhibits complex systemic endotypes characterized by immune dysregulation and hyperinflammation. Although the data obtained are valuable to the scientific community, one critique is the lack of validation of the findings in real patient tissues. It is suggested that some validation methods like Hematoxylin and Eosin staining, immunohistochemical, or immunofluorescence studies on tissues should be included, especially for potential publication in a journal like Nature Communications.

Moreover, the study's main limitation is the sample size. The wide age range of patients recruited, from 25 to 52 years old, could potentially benefit from grouping patients into smaller age ranges for analysis. Additionally, there is a question about the informativeness of the data for the clinical management of SCC in RDEB for clinicians. While the reviewer lacks a clinical background to provide a full commentary on this aspect, it is suggested that more tissue data correlated with the presented findings should be included if the authors intend to proceed with this journal submission.

Major key points that need to be addressed include:

- 1) Validation of findings in real patient tissues through staining and imaging techniques.
- 2) Grouping patients into smaller age ranges for analysis due to the wide age range in the study.
- 3) Ensuring that the data presented are informative enough for clinical management of SCC in RDEB.
- 4) Addressing the limitations of the sample size in the study.
- 5) "Recessive Dystrophic Epidermolysis Bullosa (RDEB), an incurable inherited rare skin disorder, is caused by mutations in the human COL7A1 gene encoding type VII collagen (Coll-VII), with no real cure" RDEB was described as incurable at the beginning of the sentence, and again at the end. This is not needed twice in one sentence.
- 6) "Dermal fibrosis and aggressive squamous cell carcinoma (SCC) are recurrent invalidating complications of adults RDEB". Do authors mean RDEB adults or adults with RDEB?
- 7) "The cohort included non-end-stage patients at diverse state of severity". States of severity.
- 8) Mistakes in referring to figures in text "Illustration of UMAP density plots revealed substantial differences in major innate and adaptive immune cell populations of RDEB compared to HC, particularly in neutrophils, monocytes, dendritic cells, and lymphocytes (T, B, and NK cells) (Figure 1B)" Figure 1B shows Violin plots not UMAP density plots. The UMAP density plots are labelled as 1A. The figure legends refer to the violin plots as 1C, however the figure does not have a C labelling.
- 9) "C" Violin plots comparing the frequency (%) and the absolute counts (cells/l) of indicated immune cell populations in healthy controls (black triangles) and RDEB patients (red circles)." This is the legend for Figure 2. I can only see absolute counts in 2C, but frequency % is not there as suggested by the figure legend.
- 10) Figure 3. "UMAP from healthy controls (n=9) and RDEB patients (n=12) clustered" The figure suggests this is a tSNE analysis however the legend says UMAP. These are two different algorithm analyses. Not consistent.
- 11) "RDEB patients also have an overabundance in effector CD8+ T...." Overabundance of effector CD8+.

12) Figure 3E-F. 1. Should have clear labels on the figure of which graphs represent CD4+ cells or CD8+ cells. 2. Would benefit from showing the HC and RDEB T cell proliferation with and without stimulation with CD3/CD28/CD2 antibody complexes as a control before the HC proliferation is compared to RDEB T cell proliferation.

13) "Manual gating on CD3, CD4, CD8, CD56 expressing cells combined with SCENITH allowed us to identify and analyze the cellular metabolism of T and NK cells" This statement would benefit of a supplementary figure showing the gating strategy.

14) Figure 5A. There is no clear labelling of x-axis for the histogram representation. Is this fluorescence intensity of Puromycin?

15) "Lipids, in particular bioactive mediators, act as wildfire-starters of inflammatory processes and have been clearly linked to several chronic diseases, including cancer and autoimmune disorders" This statement would benefit from a reference.

16) "Within the observed RDEB immune metabolic signature, we evaluated common pro- and anti-inflammatory cytokines in sera from RDEB patients" Mention which of the cytokine markers are referred to as pro- or anti-inflammatory.

17) "Similar scenario might occur in RDEB patients displaying high level of various pro-inflammatory cytokines" A similar scenario might occur in RDEB patients displaying high levels of various pro-inflammatory cytokines.

18) "Paraformaldehyde for 15 min at room temperature, permeabilized using Fix Perm buffer, then incubated with 125 nM Iridium DNA intercalator (Standard Bio Tools) for an overnight" Incubated with 125 nM Iridium DNA intercalator overnight.

19) "then 10⁵ PBMCs were stimulated with Immunocult CD3/CD28/CD2 (10971, StemCell, Grenoble, France) for 4 days" Detail of stimulation antibody concentration in PBMC?

Reviewer #2

(Remarks to the Author)

The manuscript describes in detail immunological and metabolic profiles in 12 adult patients with recessive dystrophic epidermolysis bullosa. Although the detailed immunological profile is valuable, the way the results are placed in the context of epidermolysis bullosa is erroneous. This starts with the "expert opinion" disseminated in review articles that dystrophic epidermolysis bullosa is an inflammatory disease and inflammation modifies the phenotype. Where the inflammation comes from remains ambiguous and is also not elucidated in this manuscript. This is the major flaw of the manuscript. It does not recognize that the immunological changes found in adult patients result from decades of large surface wounds with heavy bacterial colonisation and repeated episodes of infection. These lead to a hyperimmune response that actually manages to protect patients with EB from environmental insults and severe infections in the absence of epidermis. At some point, the immune cells, keratinocytes and fibroblasts become exhausted. In the entire manuscript, the word "wound" - the main actor in this pathogenic chain - is missing.

Altogether the manuscript remains descriptive and does not explain mechanisms. It suggests that inflammation is an additional independent feature of epidermolysis bullosa, and its treatment is an alternative approach. It does not recognize that inflammation is initially an adaptive /response mechanism secondary to skin fragility and wounding. In the end, systemic inflammation is the consequence of large wounds over a long time.

To make things more complicated, few patients in this study already had SCC.

Minor points:

1) Mutations are missing, as well as the DEB subtype.

2) iscoreEB is a broad scoring system including many items and does not discriminate between individual disease domains/manifestations. It would be more important to score individual disease manifestations and try to correlate them with immunological findings, in particular wound area.

3) The main results showing upregulated IFN, IL6, IL17 recapitulate previous results from the literature (please add such recent human studies to the references).

4) Dermal fibrosis is actually scarring

5) The first sentence should be reconsidered since it contains incurable and cure.

Reviewer #3

(Remarks to the Author)

Reviewer #4

(Remarks to the Author)

In this manuscript Hirt et al. investigate the differences in multiple aspects of immune activation in human adult patients with recessive dystrophic epidermolysis bullosa (RDEB) compared with healthy subjects. The manuscript is of interest to the field and leverages multiple state-of-the-art techniques to characterize the modifications induced by RDEB. Overall, however, the study is quite descriptive in nature. My detailed comments are below:

- As mentioned above the study seems overly descriptive. The authors do quite elegant work to demonstrate changes in number and abundance of cell types, levels of cytokines and lipids, and cellular metabolism. What seems to be lacking is

linking these observations into a coherent understanding of how these changes are impacting the progression of disease. The authors do include measures of cell activity (T cell replication/proliferation; markers of NK cell degranulation) which is a strength, however, even within these analyses it is hard to link one result to the next. For instance, with CD4+ and CD8+ T cells the authors show that the abundance of certain subtypes is changed as are elements of their cellular metabolism, but their activity (RI and PI) are not significantly different. Conversely with NK cells, again subtype abundances are different and activity is changed (degranulation markers), but there is no alteration in cellular metabolism. This may be of importance to understanding the RDEB etiology but there is a lack of interpretation of the data and in conveying how these indices may be contributing to clinical pathology.

- For the mass cytometry data in Figs 1 and 2, it is somewhat unclear how the frequency of each cell type is being determined. In Fig 1 it would be assumed that this is the frequency of all circulating cells (though it may be helpful to state this more directly in the results). For Fig 2, it is less clear what the starting population is that the frequencies are being calculated from.
- In addition to the above comment, it may be helpful to the reader to include a plot of total circulating immune cells in Fig 1.
- The deep learning prediction model results are also unclear. It seems as though combining all the end points presented in the manuscript (cell type abundance, cytokine and lipid levels) does not accurately predict a clinical phenotype. This leaves the reader with the question of the importance of these indicators in translating to patient outcomes.
- In general, the manuscript is somewhat brief in its description of analyses and interpretation of results leaving the reader to largely infer their importance.

Minor comments:

- Letter denotation of panels in Fig 1 appears to be off. Legend refers to violin plots as panel C but labeled in figure as panel B
- Some of the figure/section titles are somewhat odd (e.g. RDEB patients Inflammation Immunity)

Version 1:

Reviewer comments:

Reviewer #1

(Remarks to the Author)

Thank you for revising my comments and submitting an updated version of your manuscript.

Comment 1: This has been sufficiently addressed; however, I noticed that scale bars are missing in some images in Fig 1c. Could you please rectify this issue?

Comments 2, 3, and 4: I appreciate the responses and the incorporation of the revisions into the manuscript. I am satisfied with the answers provided.

Comments 5-19: I accept all corrections. Thank you!

I have no further comments for the authors.

Reviewer #2

(Remarks to the Author)

Thank you for considering our previous comments. The value of the findings of this study is appreciated, however, there are some concerns regarding the interpretation and the discussion. My main concern is the confusion about disseminating the opinion that treating inflammation is more important than correcting the genetic defect in RDEB. Altogether, the discussion would benefit from shortening.

1. All the participants are adults over 25 years of age in whom EB has progressed over more than 2 decades and has reached the pre-end stage. We see the effect of decades of wounds and bacterial load on the skin and mucosal membranes. Analysis of a pediatric cohort (see *Br J Dermatol.* 2020 Jun;182(6):1437-1448. doi: 10.1111/bjd.18475, *J Eur Acad Dermatol Venereol.* 2024 Feb 20.

doi: 10.1111/jdv.19898.) would have been of interest in order to capture primary and secondary processes.

2. Genetics is not clear. In a few cases, only one or no mutation was disclosed.

3. "Given that our RDEB cohort is exempt from opportunistic infections at the time of sampling, the high level of pro-inflammatory metabolites concords with the observed excessive number of neutrophils, which might be related to a continuous surge from bone marrow to the circulation as in other diseases^{49,50}." This statement eludes the fact that massive amounts of bacteria have been present in the wounds of patients for decades - this important player is neither mentioned nor discussed.

4. "While the role of Coll-VII in regulating innate immunity is well-established" - is it? Of course if you consider that losing epidermis is a defect of innate immunity. Cochlin loss of function mutations are not associated with immune defects and patients with DEB do not have increased susceptibility to infections.

Reviewer #3

(Remarks to the Author)

Reviewer #4

(Remarks to the Author)

In this revised manuscript, Hirt et al. have included new data and substantial textual revisions which have aided the reader in their interpretation of the study. This reviewer's concerns centered around underdeveloped data interpretation and methods description have been largely resolved while the descriptive nature of the manuscript remains, though the authors' inclusion of a statement of limitations for the study was appropriate and an important addition.

Version 2:

Reviewer comments:

Reviewer #3

(Remarks to the Author)

Point-by-point response to REVIEWERS

Reviewer #1 (Remarks to the Author):

“This paper is a timely and comprehensive study that applies systems immunology approaches utilizing single-cell high-dimensional techniques to capture the immune cell signatures and metabolic profiles diversity in RDEB adult patients. The patients were sampled outside of any opportunistic infection or active cancer. The study reveals the distinctive inflammation and immunity characteristics of RDEB patients, with signatures of active/effector T cells and dysfunctional natural killer cells, along with an overall pro-inflammatory lipid signature—a novel finding in the field of RDEB.

The authors suggest, through primarily computer-based analyses, that RDEB is not solely a cutaneous disease but rather exhibits complex systemic endotypes characterized by immune dysregulation and hyperinflammation. Although the data obtained are valuable to the scientific community, one critique is the lack of validation of the findings in real patient tissues. It is suggested that some validation methods like Hematoxylin and Eosin staining, immunohistochemical, or immunofluorescence studies on tissues should be included, especially for potential publication in a journal like Nature Communications.

Moreover, the study's main limitation is the sample size. The wide age range of patients recruited, from 25 to 52 years old, could potentially benefit from grouping patients into smaller age ranges for analysis. Additionally, there is a question about the informativeness of the data for the clinical management of SCC in RDEB for clinicians. While the reviewer lacks a clinical background to provide a full commentary on this aspect, it is suggested that more tissue data correlated with the presented findings should be included if the authors intend to proceed with this journal submission.”

We thank the reviewer for considering our study as timely and comprehensive, and for acknowledging the novelty of our findings in the field of RDEB and their value for the scientific community.

Major key points

1) Validation of findings in real patient tissues through staining and imaging techniques.

R1) As requested by the reviewer, we did validate our results by performing imaging mass cytometry (IMC) – Hyperion using a panel of 30 metal-conjugated antibodies, and identified the immune cell infiltrates in skin biopsies from 2 patients of the studied cohort at recruitment. These data have been added to the revised manuscript as Figure 1c and 1d, Supplementary Figure 2a and 2b. and discussed under Results and Discussion sections.

2) Grouping patients into smaller age ranges for analysis due to the wide age range in the study.

R2) All patients suffer from RDEB disease at birth but present different stages of severity that inevitably worsen with age to a degree highly related to their initial one. Nonetheless, 10 out of the 12 patients included in our cohort with a mean age of 35 years presented relatively homogenous IScorEB (with a mean at 77). Two patients present the two extremes, a 52-year-old patient suffering from localized RDEB with the lowest IScorEB at 33, and a 25-year patient with severe generalized RDEB and the most severe clinical manifestations of our cohort reflected in the highest IScorEB at 133. Grouping patients in smaller groups based on their age, as suggested by the reviewer, would result in three groups: the first includes patients in their twenties (n=2), the second includes patients in their thirties (n=7), and the last group includes patients in their forties/fifties (n=3). This would leave us with a small number in each group, which would considerably limit the statistical power. Furthermore, the three groups show considerable discrepancies in the IScorEB. Since the two extremes did not affect the global IScorEB, we have decided to include all patients in the analysis. This issue has been clarified in the revised manuscript under Results section.

3) Ensuring that the data presented are informative enough for clinical management of SCC in RDEB.

R3) The reviewer raised a valid concern about whether our study provides sufficient information for the clinical management of SCC in RDEB, a significant cause of morbidity and mortality in these patients.

While our study offers valuable insights into RDEB as not just a skin disease but also a systemic inflammatory condition, it was not specifically designed to address RDEB-associated SCC.

Our in-depth analysis, supported by AI modeling of RDEB adults immune/inflammatory profiles, divided the patients into two main groups. The first group includes 8 patients with relatively moderate to high inflammatory profile, while the second group consist of 4 patients with the highest inflammatory profile (Figure 8 in both the original and revised manuscripts). Among these four patients, two had a history of SCC at the time of sampling, while the other two did not. It is important to note that all analyses were conducted with a single blood sample taken on the day of recruitment.

In this context, although a clear correlation between inflammation/immunity and SCC cannot be definitely established, we cannot rule out the possibility the SCC-free patients at recruitment may develop this type of cancer in the future. Despite extensive research from various groups, the pathogenesis of RDEB-SCC remains complex and challenging to understand, likely due to the multifactorial nature of RDEB-SCC, where the convergence of several factors, including the inflammatory microenvironment (Santucci C. et al. *Hum Immunol* 85, 2024), contribute to the development of aggressive cancers. Within the limitations of our study, we observed high levels of the inflammatory cytokine IL6, which has previously been suggested as a risk factor for the development of aggressive SCC (Condorelli AG. et al. *Int J Mol Sci* 20, 2019). Therefore, our study suggests that lessening inflammation could be a potential strategy for better management of RDEB-SCC. This question has been discussed in the revised manuscript's Discussion section.

4) Addressing the limitations of the sample size in the study.

R4) We acknowledge the reviewer's comment, and we are fully aware of the limitations inherent to our sample size. However, RDEB is an extremely rare disease, with an estimated prevalence of one patient in two million. Given this rarity, obtaining a larger sample size of RDEB adults would have been highly challenging. Although we addressed this issue under the Discussion of our original manuscript, we have further revised the paragraph to explicitly recognize the limitations of our study, as recommended by the reviewer.

5) "Recessive Dystrophic Epidermolysis Bullosa (RDEB), an incurable inherited rare skin disorder, is caused by mutations in the human COL7A1 gene encoding type VII collagen (Coll-VII), with no real cure" RDEB was described as incurable at the beginning of the sentence, and again at the end. This is not needed twice in one sentence.

R5) The sentence was corrected in the revised manuscript as advised.

6) "Dermal fibrosis and aggressive squamous cell carcinoma (SCC) are recurrent invalidating complications of adults RDEB". Do authors mean RDEB adults or adults with RDEB?

R6) We acknowledge the typo in using adults RDEB. This has been corrected throughout the revised manuscript as RDEB adults.

7) "The cohort included non-end-stage patients at diverse state of severity". States of severity.

R7) The sentence has been corrected in the revised manuscript.

8) Mistakes in referring to figures in text "Illustration of UMAP density plots revealed substantial differences in major innate and adaptive immune cell populations of RDEB compared to HC, particularly in neutrophils, monocytes, dendritic cells, and lymphocytes (T, B, and NK cells) (Figure 1B)" Figure 1B shows Violin plots not UMAP density plots. The UMAP density plots are labelled as 1A. The figure legends refer to the violin plots as 1C, however the figure does not have a C labelling.

9) "C) Violin plots comparing the frequency (%) and the absolute counts (cells/ μ l) of indicated immune cell populations in healthy controls (black triangles) and RDEB patients (red circles)." This is the legend for Figure 2. I can only see absolute counts in 2C, but frequency % is not there as suggested by the figure legend.

10) Figure 3. "UMAP from healthy controls (n=9) and RDEB patients (n=12) clustered" The figure suggests this is a tSNE analysis however the legend says UMAP. These are two different algorithm analyses. Not consistent.

11) "RDEB patients also have an overabundance in effector CD8+ T...." Overabundance of

effector CD8+.

R8, 9, 10, 11) We apologize for these unintentional typos that crept into the original manuscript. All Figures, Figures legends, and the text have been carefully reviewed and corrected, for the revised manuscript.

12) Figure 3E-F. 1. Should have clear labels on the figure of which graphs represent CD4+ cells or CD8+ cells. 2. Would benefit from showing the HC and RDEB T cell proliferation with and without stimulation with CD3/CD28/CD2 antibody complexes as a control before the HC proliferation is compared to RDEB T cell proliferation.

R12) The labelling in Figure 3E-F has been clarified as advised, and results regarding healthy donors and RDEB T cell proliferation without stimulation were added to the revised manuscript as Supplementary Figure 6.

13) “Manual gating on CD3, CD4, CD8, CD56 expressing cells combined with SCENITH allowed us to identify and analyze the cellular metabolism of T and NK cells” This statement would benefit of a supplementary figure showing the gating strategy.

R13) The manual gating on CD3, CD4, CD8, CD56 expressing cells combined with SCENITH has been added to supplementary materials as advised (see Supplementary Figure 8).

14) Figure 5A. There is no clear labelling of x-axis for the histogram representation. Is this fluorescence intensity of Puromycin?

R14) Figure 5A has been clearly labelled, with the x-axis presenting fluorescence intensity of Puromycin.

15) “Lipids, in particular bioactive mediators, act as wildfire-starters of inflammatory processes and have been clearly linked to several chronic diseases, including cancer and autoimmune disorders” This statement would benefit from a reference.

R15) Appropriate reference has been added to support the statement as recommended.

16) “Within the observed RDEB immune metabolic signature, we evaluated common pro- and anti-inflammatory cytokines in sera from RDEB patients” Mention which of the cytokine markers are referred to as pro- or anti-inflammatory.

17) “Similar scenario might occur in RDEB patients displaying high level of various pro-inflammatory cytokines” A similar scenario might occur in RDEB patients displaying high levels of various pro-inflammatory cytokines.

18) “Paraformaldehyde for 15 min at room temperature, permeabilized using Fix Perm buffer, then incubated with 125 nM Iridium DNA intercalator (Standard Bio Tools) for an overnight” Incubated with 125 nM Iridium DNA intercalator overnight.

R16, 17, 18) Pro- and anti-inflammatory cytokines have been specified in Results section as advised, and both sentences referred to in 17 and 18 have been corrected.

19) “then 10⁵ PBMCs were stimulated with Immunocult CD3/CD28/CD2 (10971, StemCell, Grenoble, France) for 4 days” Detail of stimulation antibody concentration in PBMC?

R19) The concentration of stimulation antibody has been specified in Material and Methods section.

Reviewer #2 (Remarks to the Author):

The manuscript describes in detail immunological and metabolic profiles in 12 adult patients with recessive dystrophic epidermolysis bullosa. Although the detailed immunological profile is valuable, the way the results are placed in the context of epidermolysis bullosa is erroneous. This starts with the "expert opinion" disseminated in review articles that dystrophic epidermolysis bullosa is an inflammatory disease and inflammation modifies the phenotype. Where the inflammation comes from remains ambiguous and is also not elucidated in this manuscript. This is the major flaw of the manuscript. It does not recognize that the immunological changes found in adult patients result from decades of large surface wounds with heavy bacterial colonisation and repeated episodes of infection. These lead to a hyperimmune response that actually manages to protect patients with EB from environmental insults and severe infections in the absence of epidermis. At some point, the immune cells, keratinocytes and fibroblasts become exhausted. In the entire manuscript, the word "wound" - the main actor in this pathogenic chain - is missing!

Altogether the manuscript remains descriptive and does not explain mechanisms. It suggests that inflammation is an additional independent feature of epidermolysis bullosa, and its treatment is an alternative approach. It does not recognize that inflammation is initially an adaptive /response mechanism secondary to skin fragility and wounding. In the end, systemic inflammation is the consequence of large wounds over a long time. To make things more complicated, few patients in this study already had SCC!

Major point:

We appreciate the reviewer's recognition of the value in our in-depth immunological and metabolic characterization of RDEB patients. However, we also understand their concerns regarding the inclusion of the immune system and inflammation in the context of RDEB.

Response to major point:

Accumulating evidence indicates that RDEB involves intrinsic dysregulation of the systemic immune system, advancing our understanding of RDEB as not only a skin disease but also a systemic inflammatory condition. We apologize if our wording was ambiguous, leading to the reviewer's concerns. Our manuscript does not claim that inflammation acts as an independent feature modifying the phenotype. Instead, it describes the inflammation-immunity character of RDEB adults and suggests that reducing inflammation and restoring appropriate immune response can improve their wound healing and life conditions, presenting an approach focused on symptom relief.

The source of inflammation in RDEB remains an important and intriguing question, comparable to the age-old riddle: "Which came first, the hen or the egg?". Our study was not designed to, and therefore cannot provide conclusive mechanistic insights into the origin of inflammation. We fully agree with the reviewer that the immunological changes observed in adult patients may indeed result from decades of extensive skin wounds and recurrent infections. These factors could trigger a hyperimmune response aimed at protecting EB patients from environmental insults and severe infections in the absence of an intact epidermis. However, wound-related hyperimmune activation may not be the only possible explanation. The RDEB adults in our cohort exhibit varying percentages of wound areas, ranging from 20% to 80%. Notably, 4 out of 12 patients with the highest inflammatory signatures did not consistently present the highest wound percentages (see revised Figure 8b and Supplementary Table 2). This suggests that the systemic inflammation in RDEB may not be exclusively a consequence of extensive wounds over time, and may originate from other factors beyond skin wounds.

A large body of evidence indicates that RDEB involves intrinsic dysregulation of the systemic immune system. Notably, the work by Nystrom A. et al. has demonstrated the role of the collagen VII-cochlin axis in lymphoid organs in regulating innate immunity, supporting the notion of an intrinsic innate immune dysfunction in RDEB. This observation has shifted the perception of RDEB from being solely a mucocutaneous blistering disorder to a systemic, multiorgan disease (Nystrom A. et al, *Proc Natl Acad Sci U S A* 115, 2018). Although the impact of dysfunctional collagen VII on adaptive immunity is not yet fully understood, collagen VII is present in thymic basement membrane (Virtanen I, et al. *Histochem J* 28, 1996) and secondary lymphoid organs, namely lymph nodes and spleen (Nystrom A. et al, *Proc Natl Acad Sci U S A* 115, 2018; Huitema L, *Exp Dermatol* 30, 2021). These findings suggest that dysfunctional collagen VII may also influence adaptive immunity. Our study demonstrates a distinct T and NK cell signature and function in RDEB patients, aligning with the above discussed concepts. We fully agree that wounds have a central role in the RDEB pathological chain, and our intention was not to

exclude them or diminish their importance but to provide an in-depth analysis of the peripheral immune landscape of RDEB adults, a topic that has not been fully explored in the literature.

To address the reviewer's concern, we have included in the revised manuscript the percentage of wound area of our cohort (Supplementary Table 2), and provided a snapshot of the wound immune landscape using biopsies taken at patient enrollment and the time of blood sampling, outside of any opportunistic infection. Imaging mass cytometry (IMC) analysis showed dynamic changes in the proportions of various immune cell populations in skin biopsies from two RDEB adults in the studied cohort. These changes were consistent with those observed in peripheral blood immune cells. In RDEB mouse model, a time-dependent increase in adaptive T cell infiltrates, with a shift from activation to that of exhaustion occurs and was correlated with progressive fibrosis. Similar findings were observed in human RDEB skin from different stages of fibrosis/disease (Ebens CL. *EMBO Mol Med* 13, 2021). Our IMC analysis also showed remarkably high levels of activated CD4⁺ and CD8⁺ T cells infiltrates that co-localise with fibrotic region in skin lesions from RDEB adults, further emphasizing the connection between systemic immune dysregulation and disease pathophysiology. These data have been added to the Results section and are depicted in Panel c and d of the revised Figure 1, and supplementary Figure 2a,b. Therefore, systemic inflammation in RDEB may not be solely a consequence of extensive wounds over time, and may originate from other factors beyond skin wounds.

Nine out of twelve patients in our cohort had a history of SCC at the time of enrollment. Although we do not fully understand the reviewer's point in stating, "To make things more complicated, few patients in this study already had SCC!", we did not intend to exclude this important cause of morbidity and mortality in RDEB adults. Our study provides valuable insights into understanding RDEB as not just a skin disease but also a systemic inflammatory condition. However, it was not designed to address specifically RDEB-SCC. Our in-depth analysis, supported by AI modeling of patients' immune/inflammatory components, divided them into two main groups: one comprising eight patients with a relatively moderate to high inflammatory profile, and another including 4 patients with the highest inflammatory profile (Figure 8 in both the original and revised manuscripts). Among these 4 patients, two had a history of SCC at the time of sampling, while the other two did not. It is important to note that all the analyses were conducted with a single blood sample taken the day of recruitment. In this context, although we cannot draw a definitive correlation between inflammation/immunity and SCC, we cannot exclude that SCC-free patients at recruitment may not develop this type of cancer in the future. Despite the extensive research from various groups, understanding of the pathogenesis of RDEB-SCC remains challenging because of its multifactorial nature, where the convergence of several factors, including inflammatory microenvironment (Santucci C. et al. *Hum Immunol* 85, 2024), appears to drive the development of aggressive cancer. Within the limits of our study, we observed a high level of the inflammatory cytokine IL6, which has been previously suggested as a risk factor for the development of aggressive SCC (Condorelli AG. et al. *Int J Mol Sci* 20, 2019). Based on these findings, our study suggests that lessening inflammation may enable a better management of RDEB-SCC.

We have incorporated part of our response to the major concern raised by Reviewer 2 into the Results and Discussion sections of the revised manuscript.

Minor points:

1) Mutations are missing, as well as the DEB subtype.

R1) A table with all mutations has been added to the revised manuscript as Supplementary Table 1.

2) iscorEB is a broad scoring system including many items and does not discriminate between individual disease domains/manifestations. It would be more important to score individual disease manifestations and try to correlate them with immunological findings, in particular wound area.

R2) While we acknowledge the reviewer's concern that the IScorEB might not be the ideal way to score patients, it is the standard scoring method used by the French National Reference Centre for Rare Diseases of the Skin and Mucous Membranes of Genetic Origin (MAGEC, which manages the RDEB adults cohort). Therefore, we chose to keep it in our original predictions and correlations. However, to address the reviewer's concerns, we refined our predictions and PCA analysis to include the percentage of wound area for each patient (included in Supplementary Table 2 of the revised manuscript) as well as the SCC history as well. These updated predictions are presented in the revised Figure 8 and are discussed under Results section.

3) The main results showing upregulated IFN, IL6, IL17 recapitulate previous results from the literature (please add such recent human studies to the references).

R3) Although we cited in our submitted manuscript, we may have unintentionally missed some. We have revised the manuscript to include the missing references, along with a recent review paper that summarizes all previous findings on this topic (Santucci C, et al. *Hum Immunol* 85, 2024).

4) Dermal fibrosis is actually scarring

R4) We understand that dermal fibrosis, a significant burden for RDEB patients, is often associated with scarring. However, it is now recognized that fibrosis is directly correlated with collagen VII abnormalities and is partially independent of the healing process. Indeed, many areas of skin fibrosis develop in the absence of preceding wounds. For instance, synechiae on the hands and feet can occur without prior wounds. Therefore, dermal fibrosis might not always mirror scarring.

5) The first sentence should be reconsidered since it contains incurable and cure.

R5) The first sentence has been revised as follow “Recessive Dystrophic Epidermolysis Bullosa (RDEB), an inherited rare skin disorder with no real cure, is caused by mutations in the human *COL7A1* gene encoding type VII collagen (Coll-VII)”

Reviewer #3 (Remarks to the Author):

We thank the reviewer for their time and effort.

Reviewer #4 (Remarks to the Author):

“In this manuscript Hirt et al. investigate the differences in multiple aspects of immune activation in human adult patients with recessive dystrophic epidermolysis bullosa (RDEB) compared with healthy subjects. The manuscript is of interest to the field and leverages multiple state-of-the-art techniques to characterize the modifications induced by RDEB. Overall, however, the study is quite descriptive in nature.”

We appreciate that the reviewer found our manuscript of interest to the field and that they acknowledged the in-depth leverage of multiple state-of-the-art techniques to characterize the modifications induced by RDEB. However, they raised some concerns that we address below.

1) As mentioned above the study seems overly descriptive. The authors do quite elegant work to demonstrate changes in number and abundance of cell types, levels of cytokines and lipids, and cellular metabolism. What seems to be lacking is linking these observations into a coherent understanding of how these changes are impacting the progression of disease. The authors do include measures of cell activity (T cell replication/proliferation; markers of NK cell degranulation) which is a strength, however, even within these analyses it is hard to link one result to the next. For instance, with CD4+ and CD8+ T cells the authors show that the abundance of certain subtypes is changed as are elements of their cellular metabolism, but their activity (RI and PI) are not significantly different. Conversely with NK cells, again subtype abundances are different and activity is changed (degranulation markers), but there is no alteration in cellular metabolism. This may be of importance to understanding the RDEB etiology but there is a lack of interpretation of the data and in conveying how these indices may be contributing to clinical pathology.

R1) We fully understand the reviewer's perspective on integrating our in-depth analysis of the immune-inflammatory signature with the progression of RDEB. Recognizing the limitations of our study, we acknowledge that we might have been overly conservative in placing our results within the context of RDEB progression.

Mouse studies and some reports in human settings highlighted the immune/inflammatory mechanisms as a modifier of RDEB progression. Within this notion, our in-depth analyses emphasized and furthered the understanding of the immune-inflammatory features in human RDEB. The immune system relies on a balanced array of homeostatic mechanisms to manage various insults. Disruption of this delicate equilibrium often leads to pathological disorders.

Developing a given immune cell phenotype requires appropriate interactions between cell metabolic states, signaling, and transcription responses to ensure the availability of building molecules and energy for cellular homeostasis. In line with their metabolic adaptations, RDEB CD4 and CD8 T cell compartments show disturbed homeostasis of naive and effector cell subtypes, likely biased toward an active/effector phenotype. Furthermore, similar to chronic infection and exhaustion, alterations in the NK cell compartment are characterized by an increased proportion of CD57-positive cells and impaired cytotoxicity, likely due to metabolic reprogramming towards decreased translation. Noteworthy, Imaging mass cytometry analysis of 2 skin biopsies taken from 2 patients at enrollment and the time of blood sampling, outside of any opportunistic infection showed dynamic changes in the proportions of various immune cell populations that reflected those observed with peripheral immune cells. These data are added to Results section and depicted in Panel c and d of the revised Figure 1, and supplementary Figure 2a, b. Thus, metabolism-dependent disruptions of immune cell phenotype and function affect both peripheral T and NK cell compartments, and might nurture a main actor of pathologic chain of RDEB, unhealed chronic skin wounding.

Bioactive lipid mediators, which act as wildfire-starters of inflammatory processes, have been linked to inflammation and cancer. Hence, the pro-inflammatory lipid signature of our RDEB cohort fosters the inflammatory microenvironment of these patients, shifting their immune compartment metabolism towards active/effector T cells and concomitant dysfunctional NK cells. Consequently, the metabolic alterations in RDEB that imprint various immune components, would create a conducive environment for infections and cancer. The convergence of several factors, including an inflammatory microenvironment, currently presents a scenario promoting the development of SCC, which represents a significant cause of morbidity-mortality in RDEB patients. The observed immune-metabolic alterations

and persistent inflammatory microenvironment might, therefore, contribute to the progression of SCC in RDEB.

Besides the added results, part of these clarifications has been included in the Discussion section of the revised manuscript to illustrate our perception.

2) For the mass cytometry data in Figs 1 and 2, it is somewhat unclear how the frequency of each cell type is being determined. In Fig 1 it would be assumed that this is the frequency of all circulating cells (though it may be helpful to state this more directly in the results). For Fig 2, it is less clear what the starting population is that the frequencies are being calculated from.

R2) As recommended by the reviewer, the starting population has been added to the results describing Figures 1 and 2, and Supplementary Figure 1a, as well as to respective Material and Methods sections.

3) In addition to the above comment, it may be helpful to the reader to include a plot of total circulating immune cells in Fig 1.

R3) We agree with the reviewer's comment. For mass cytometry and according to the manufacturer protocol, we start with an identical number of total immune circulating cells to perform the acquisition. A fixed number of 500,000 CD45-positive circulating immune cells were acquired for each patient and control to avoid any eventual bias in the analyses. The gating strategy has been further clarified for the starting population (see the revised Supplementary Figure 1a). These clarifications are now included in respective Material and Methods sections and legends to figures.

4) The deep learning prediction model results are also unclear. It seems as though combining all the end points presented in the manuscript (cell type abundance, cytokine and lipid levels) does not accurately predict a clinical phenotype. This leaves the reader with the question of the importance of these indicators in translating to patient outcomes.

R4) We thank the reviewer for bringing this to our attention. Our original intention was not to predict a particular clinical phenotype but rather to evaluate the importance of the immune-inflammatory character of adults RDEB.

Our prediction model underscores the importance of including the immune-inflammatory status as an assessment parameter of disease to refine the IscorEB to ultimately reach better patient management. To highlight the significance of our study, part of this clarification has been included in the discussion section of the revised manuscript.

5) In general, the manuscript is somewhat brief in its description of analyses and interpretation of results leaving the reader to largely infer their importance.

R5) We appreciate the comments from the reviewer, which helped us improving and extending our results and their integration within the pathophysiology of RDEB. Discussion section of the revised manuscript illustrates our perception in a clearer manner.

Minor comments:

1) letter denotation of panels in Fig 1 appears to be off. Legend refers to violin plots as panel C but labeled in figure as panel B

-This has been corrected.

2) Some of the figure/section titles are somewhat odd (e.g. RDEB patients Inflammation Immunity)

-We considered the reviewer point of view, and replaced the results section title "RDEB Inflammation Immunity" by "Lipid and immunometabolism interconnectivity in RDEB", and the title of Figure 7 "RDEB adults Inflammation Immunity" by "Mapping lipid rewiring in RDEB" in the revised manuscript.

Response to Reviewers

Reviewer #1 (Remarks to the Author):

Thank you for revising my comments and submitting an updated version of your manuscript.

Comment 1: This has been sufficiently addressed; however, I noticed that scale bars are missing in some images in Fig 1c. Could you please rectify this issue?

As suggested by the reviewer we included the scale bar in the enlargement micrograph in Fig. 1c.

Comments 2, 3, and 4: I appreciate the responses and the incorporation of the revisions into the manuscript. I am satisfied with the answers provided.

Comments 5-19: I accept all corrections. Thank you!

I have no further comments for the authors.

We sincerely appreciate the reviewer comments and suggestions that largely improved our manuscript.

Reviewer #2 (Remarks to the Author):

Thank you for considering our previous comments. The value of the findings of this study is appreciated, however, there are some concerns regarding the interpretation and the discussion. My main concern is the confusion about disseminating the opinion that treating inflammation is more important than correcting the genetic defect in RDEB. Altogether, the discussion would benefit from shortening.

We apologize if the reviewer got the impression that we are disseminating the opinion that lessening inflammation is more important than genetic corrections. By no means we intended to diminish the importance of the only curative treatment for patients, the gene therapy. To avoid any possible misleading statements, we replaced the wording in the abstract and the discussion as you can see below

Abstract, "By revealing the phenotype-endotype association in RDEB adults, our study lay the groundwork for translational interventions that could by lessening inflammation, alleviate the everlasting suffering of RDEB patients, while awaiting curative genetic therapies."

Discussion, "While awaiting full clinical translation of *COL7A1* genetic corrections, our artificial intelligence interrogation, which emphasized the phenotype-endotype association in RDEB adults, suggests that monitoring of immune and inflammatory metabolic states could improve the management of these patients."

1. All the participants are adults over 25 years of age in whom EB has progressed over more than 2 decades and has reached the pre-end stage. We see the effect of decades of wounds and bacterial load on the skin and mucosal membranes. Analysis of a pediatric cohort (see Br J Dermatol. 2020 Jun;182(6):1437-1448. doi: 10.1111/bjd.18475, J Eur Acad Dermatol Venereol. 2024 Feb 20.doi: 10.1111/jdv.19898.) would have been of interest in order to capture primary and secondary processes.

We understand the reviewer point of view and agree that only larger retrospective pediatric and adult cohorts can provide the full picture. This has been carefully further discussed under Discussion section in the limitations of the study.

2. Genetics is not clear. In a few cases, only one or no mutation was disclosed.

We appreciate that the reviewer brought this to our attention, but we reported all the genetic mutations that we have in our hand, which were contributed by Alain Hovnanian who

supervised the genetic analysis of *COL7A1* of these patients. The only information missing is for patient (IHB706). However, to be clearer in this regard, we added the phrase "Patients included in our study were diagnosed with *COL7A1* variants that led to gene splicing, frameshift, or introduction of early stop codons." in regard of genetics to the discussion section.

3. "Given that our RDEB cohort is exempt from opportunistic infections at the time of sampling, the high level of pro-inflammatory metabolites concords with the observed excessive number of neutrophils, which might be related to a continuous surge from bone marrow to the circulation as in other diseases^{49,50}." This statement eludes the fact that massive amounts of bacteria have been present in the wounds of patients for decades - this important player is neither mentioned nor discussed.

We thank the reviewer for bringing this to our attention and apologize if it has not been clearly discussed. We included to the discussion section the following paragraph "Given that our RDEB cohort was free from opportunistic infections at the time of sampling, the elevated levels of pro-inflammatory metabolites concord with the observed excessive neutrophils counts. While we cannot rule out the possibility of prior infections contributing to this, the increased neutrophil number in our cohort might also be related to a continuous surge from bone marrow into the circulation, similar to what is observed in other diseases."

4. "While the role of Coll-VII in regulating innate immunity is well-established" - is it? Of course if you consider that losing epidermis is a defect of innate immunity. Cochlin loss of function mutations are not associated with immune defects and patients with DEB do not have increased susceptibility to infections.

We agree with the reviewer that using the "well-established" might be misleading and raise confusions. To clarify this, we changed the wording in a more comprehensive manner that reflects our perception. This has been clarified under discussion section as "While a role for Coll-VII in regulating innate immunity has been put forward, its potential involvement in adaptive immunity has yet to be explored. Nonetheless, the presence of Coll-VII in the thymus basement membrane and secondary lymphoid organs suggests a possible role in regulating adaptive immune cells.

Reviewer #3 (Remarks to the Author):

detailed

ells with decreased degranulation capacity and global activity, T cells with higher translational levels and high metabolic activity, different expression profiles of eicosanoids and other lipids, and increased levels of IFN γ , TNF α , IL-17A, IL-6, and IL-10. They further associate these healthy

I have the following major comments:

-lines 103-104: healthy young controls that are blood donors are included, so one would expect that these controls do not suffer at the moment of blood sampling from any infections or autoimmune diseases, or cancer. However, it is per se expected that if RDEB patients, which are basically lacking their skin barrier, are compared with completely healthy controls, they will demonstrate a dysregulated/hyperactive immune system, with more neutrophilic activation and increased CD4 and CD8 cells. In order to purely understand the unique pathophysiology of DEB and how the immune dysregulation is different in DEB when compared to other conditions, DEB patients should be compared with milder forms of DEB or other forms of EB,

or even other forms of skin detachment diseases, like for example burns, not with completely healthy individuals that have an intact skin barrier.

The aim of our study is to provide a comprehensive understanding of the immunometabolic landscape in RDEB adults. Given this aim, patients were compared to age-matched healthy donors. The comparison of RDEB to other forms of EB or skin detachment diseases is certainly of interest in order to define a unique signature that can discriminate the different pathologies, and used as a diagnostic tool. However, this falls far beyond the scope of our actual study.

-lines 147-148: these lines belong to the discussion; results should not be analyzed in the results section. Here the authors mention that the immune system homeostasis is disturbed in DEB but in the results above no disruption is described, but rather an increased activation of the immune system (increased number of neutrophils, B cells and plasma cells, as well as CD4 and CD8), so that this conclusion cannot be drawn from the results presented. It has not been demonstrated that these cells are malfunctioning. And an increased activation of the immune system in DEB is anyhow awaited, as the skin barrier does not function.

We thank the reviewer for bringing this to our attention, the conclusion of the paragraph has been replaced by "Taken together, these results show that RDEB adults display different proportions of various immune cell populations both in the periphery and the skin".

-the authors report in the result section percentages of the immune cells and make comparisons, that not always have the same statistical significance with the comparisons made for absolute cell numbers. In my opinion only absolute cell numbers and comparisons of them should be reported and not percentages.

Because frequency and absolute numbers do not fully elude the same information, we choose to show and compare both in order to provide full clear image of the immunological landscape in the periphery.

-lines 197-199 and 206-208: these lines are a repetition and again here conclusions should not be included in the results section. They belong to the discussion. Moreover, lines 206-208 have nothing to do with the results paragraph they belong to.

We thank the review for bringing this to our attention, repetitions are deleted and the conclusion of the paragraph is corrected as "Overall, these results show that RDEB nurtures irregular ratio of naïve to effector T cells in both CD4+ or CD8+ compartments likely biased towards an effector profile"

-lines 259-260: "RDEB NK cells harbored lower glucose and mitochondrial dependence concomitant with lower translation levels" these results are not statistically significant, so you cannot report that the levels are lower (might be by chance). Either not report the results or only mention that there was no statistically significant difference in these levels.

The description was corrected to the following "The glycolytic and FAO capacity of NK cells in RDEB patients were comparable to HC. Although not statistically significant, RDEB NK cells showed a tendency to reduce their glucose and mitochondrial dependence as well as translation levels (Fig. 5d)."

-lines 329-331: the high IIS was correlated inflammatory and antiinflammatory cytokines, lipid mediators but not with the most clinical parameters like IscorEB, % wound area, or incidence of SCC. How do you explain that? How can increase IIS not correlate with the wound area? This should be discussed. Does it correlate with other important clinical parameters, like bacterial swabs, systemic or chronic infections, systemic manifestations of the disease, chronic wounds vs acute wounds?

-lines 339-340: the authors here state that the monitoring of inflammatory states might help improve the clinical management of patients. How would this be? What is the correlation of the results of this paper with the clinical image & clinical manifestations of the patients? How would the monitoring of the inflammatory state improve the clinical management of these complex patients and what can it change the management they are already receiving? This is generally unclear throughout the manuscript, that is to say the translational impact of the results.

The Clinico-pathophysiology of RDEB is highly complex. The only established and recognized parameter of RDEB is indeed the genetic variants, the primary cause of blistering and wounds. Whether and how additional parameters, if any, contribute to the progression of the disease, is not yet fully understood.

This pioneer merging of the immunological, inflammatory, and metabolic status of adult RDEB patients highlighted that IIS (peripheral cytokine levels, lipid profiles, and absolute counts of circulating immune cells) might constitute a parameter of the complex clinico-pathophysiology of this devastating disease. This notion is the aim of our study and is fully reflected in our title. Defining endotypes is important for management and standard care procedures for various diseases with complex etiology such as allergic diseases.

This being clarified, we were not aiming to establish any correlation of the IIS with the wounds, whether acute or chronic. Sure enough, our AI prediction models did not reveal any correlation. Thus, our study suggests to include this accessible systemic parameter as additional and complementary to the IScorEB for a better management of these suffering patients. Unfortunately, it is limited by the small number that did not allow for patient stratification, and calls upon gathering the efforts to investigate larger retrospective pediatric and adult cohorts to capture the full picture of the disease and sort out this notion.

Part of the above details has been included in the Discussion section of the revised manuscript.

-lines 342-344, 347-350: again, here this overabundance of immune cells is a part of the chronic inflammation that is awaited and expected in these patients that literally lack their skin barrier. No unexpected immune dysregulation is reported here, that could have also been the target of a systemic therapy. The authors should be careful not to confuse the words immune dysregulation and immune hyperactivation, their results highlight more or less a hyperactivation, than a dysregulation.

We do understand the reviewer confusions, due to a misuse of the word “dysregulation”. The statement has been corrected as “The IMC analysis showing remarkable high levels of activated CD4⁺ and CD8⁺ T cell infiltration co-localising with fibrotic region in skin lesions of RDEB adults, further support these findings and underscore the interrelation between systemic immunity and disease pathophysiology”.

-lines 390-391: the reference is outdated and obsolete, we now know that DEB is a genetic disease due to lack of Collagen 7

Although we do not fully understand what the reviewer means, the reference even if it is outdated it reported the relevance of AA in RDEB patients, and it was cited in this regard.

-lines 453-455: “systemic strategies that aim to regulate the immune system homeostasis to lessen inflammation without compromising immune defense”: again, here it is unclear, do the results and this paper speak about dysregulated inflammatory profile, or hyperactivation of the immune system? And if there is a dysregulation, which is it and how can it be pharmaceutically targeted?

We understand why the word “regulating” might have led to confusion therefore, we replaced it by a more precise word “fine-tuning” to reflect our perception.

Minor comments

-lines 110-118, 134-137, 173-178, 243-251: they all belong in the methods section, not in the results

We do understand the reviewer point of view. However, it is very common to give a brief description of methods within the result sections to facilitate the understanding of the results and the related figures. Therefore, we choose to keep these brief descriptions within the results section.

-lines 307-312: these lines belong to the discussion.

Lines 307-312 describe data in figure 7 and therefore, they belong to the result section of the manuscript.

Finally, we thank the reviewer for her/his thorough consideration of the manuscript and for pinpointing details that certainly ameliorated the manuscript to better reflect our scope and perception.

Reviewer #4 (Remarks to the Author):

In this revised manuscript, Hirt et al. have included new data and substantial textual revisions which have aided the reader in their interpretation of the study. This reviewer's concerns centered around underdeveloped data interpretation and methods description have been largely resolved while the descriptive nature of the manuscript remains, though the authors' inclusion of a statement of limitations for the study was appropriate and an important addition.

We thank the reviewer for their valuable comments that made the manuscript flow better.

d

ells with decreased degranulation capacity and global activity, T cells with higher translational levels and high metabolic activity, different expression profiles of eicosanoids and other lipids, and increased levels of IFN γ , TNF α , IL-17A, IL-6, and IL-10. They further associate these

i

l

b

d

have the following major comments:

lines 103-104: healthy young controls that are blood donors are included, so one would expect that these controls do not suffer at the moment of blood sampling from any infections or autoimmune diseases, or cancer. However, it is per se expected that if RDEB patients, which are basically lacking their skin barrier, are compared with completely healthy controls, they will demonstrate a dysregulated/hyperactive immune system, with more neutrophilic activation and increased CD4 and CD8 cells. In order to purely understand the unique pathophysiology of DEB and how the immune dysregulation is different in DEB when compared to other conditions, DEB patients should be compared with milder forms of DEB or other forms of EB, or even other forms of skin detachment diseases, like for example burns, not with completely healthy individuals that have an intact skin barrier.

-lines 147-148: these lines belong to the discussion; results should not be analyzed in the results section. Here the authors mention that the immune system homeostasis is disturbed in DEB but in the results above no disruption is described, but rather an increased activation of the immune system (increased number of neutrophils, B cells and plasma cells, as well as CD4 and CD8), so that this conclusion cannot be drawn from the results presented. It has not been demonstrated that these cells are malfunctioning. And an increased activation of the immune system in DEB is anyhow awaited, as the skin barrier does not function.

-the authors report in the result section percentages of the immune cells and make comparisons, that not always have the same statistical significance with the comparisons made for absolute cell numbers. In my opinion only absolute cell numbers and comparisons of them should be reported and not percentages.

-lines 197-199 and 206-208: these lines are a repetition and again here conclusions should not be included in the results section. They belong to the discussion. Moreover, lines 206-208 have nothing to do with the results paragraph they belong to.

-lines 259-260: "RDEB NK cells harbored lower glucose and mitochondrial dependence concomitant with lower translation levels" these results are not statistically significant, so you cannot report that the levels are lower (might be by chance). Either not report the results or only mention that there was no statistically significant difference in these levels.

-lines 329-331: the high IIS was correlated inflammatory and antiinflammatory cytokines, lipid mediators but not with the most clinical parameters like IscorEB, % wound area, or incidence of SCC. How do you explain that? How can increase IIS not correlate with the wound area? This should be discussed. Does it correlate with other important clinical parameters, like bacterial swabs, systemic or chronic infections, systemic manifestations of the disease, chronic wounds vs acute wounds?

-lines 339-340: the authors here state that the monitoring of inflammatory states might help improve the clinical management of patients. How would this be? What is the correlation of

the results of this paper with the clinical image & clinical manifestations of the patients? How would the monitoring of the inflammatory state improve the clinical management of these complex patients and what can it change the management they are already receiving? This is generally unclear throughout the manuscript, that is to say the translational impact of the results.

-lines 342-344, 347-350: again, here this overabundance of immune cells is a part of the chronic inflammation that is awaited and expected in these patients that literally lack their skin barrier. No unexpected immune dysregulation is reported here, that could have also been the target of a systemic therapy. The authors should be careful not to confuse the words immune dysregulation and immune hyperactivation, their results highlight more or less a hyperactivation, than a dysregulation.

-lines 390-391: the reference is outdated and obsolete, we now know that DEB is a genetic disease due to lack of Collagen 7

-lines 453-455: "systemic strategies that aim to regulate the immune system homeostasis to lessen inflammation without compromising immune defense": again here it is unclear, do the results and this paper speak about dysregulated inflammatory profile, or hyperactivation of the immune system? And if there is a dysregulation, which is it and how can it be pharmaceutically targeted?

Minor comments

-lines 110-118, 134-137, 173-178, 243-251: they all belong in the methods section, not in the results

-lines 307-312: these lines belong to the discussion.